# Myeloid lineage enhancers drive oncogene synergy in CEBPA/CSF3R mutant acute myeloid leukemia

Theodore P. Braun [1,2], Mariam Okhovat[3], Cody Coblentz[1,2], Sarah A. Carratt [1,2], Amy Foley[1,2], Zachary Schonrock[1,2], Brittany M. Curtiss [1,2], Kimberly Nevonen [3], Brett Davis [3], Brianna Garcia[1,2], Dorian LaTocha[1,2], Benjamin R. Weeder [4], Michal R. Grzadkowski[5], Joey C. Estabrook[5], Hannah G. Manning[5], Kevin Watanabe-Smith[1,5], Sophia Jeng[6], Jenny L. Smith[7], Amanda R. Leonti[7], Rhonda E. Ries[7], Shannon McWeeney [1,6], Cristina Di Genua[8], Roy Drissen [8], Claus Nerlov [8], Soheil Meshinchi[7,9], Lucia Carbone[3,6,10], Brian J. Druker [1,2,11] & Julia E. Maxson[1,2]*

Acute Myeloid Leukemia (AML) develops due to the acquisition of mutations from multiple functional classes. Here, we demonstrate that activating mutations in the granulocyte colony stimulating factor receptor (CSF3R), cooperate with loss of function mutations in the transcription factor CEBPA to promote acute leukemia development. The interaction between these distinct classes of mutations occurs at the level of myeloid lineage enhancers where mutant CEBPA prevents activation of a subset of differentiation associated enhancers. To confirm this enhancer-dependent mechanism, we demonstrate that CEBPA mutations must occur as the initial event in AML initiation. This improved mechanistic understanding will facilitate therapeutic development targeting the intersection of oncogene cooperativity.

[1] Knight Cancer Institute, Oregon Health & Science University, Portland, OR 97239, USA. [2] Division of Hematology and Medical Oncology, Oregon Health & Science University, Portland, OR 97239, USA. [3] Knight Cardiovascular Institute, Oregon Health & Science University, Portland, OR 97239, USA. [4] Program in Molecular and Cellular Biology, Oregon Health & Science University, Portland, OR 97239, USA. [5] Computational Biology Program, Oregon Health & Science University, Portland, OR 97239, USA. [6] Division of Bioinformatics and Computational Biology, Oregon Health & Science University, Portland, OR 97239, USA. [7] Clinical Research Division, Fred Hutchinson Cancer Research Center, 1100 Fairview Ave N, Seattle, WA 98109, USA. [8] MRC Molecular Haematology Unit, MRC Weatherall Institute of Molecular Medicine, John Radcliffe Hospital, Headley Way, Oxford OX3 9DS, UK. [9] Division of Pediatric Hematology/Oncology, University of Washington, Seattle, WA 98195, USA. [10] Department of Molecular and Medical Genetics, Oregon Health & Science University, Portland, OR 97239, USA. [11] Howard Hughes Medical Institute, Portland, OR, USA. *email: maxsonj@ohsu.edu

A cute myeloid leukemia (AML) is a deadly hematologic malignancy that results from the stepwise acquisition of genetic aberrations. The mutations that collaborate to produce AML often occur in distinct functional categories[1]. Class I mutations activate signaling pathways and in isolation produce disease with a myeloproliferative phenotype. Class II mutations perturb the function of transcription factors or epigenetic modifiers—and alone can cause a myelodysplastic phenotype. However, when present in combination, Class I and Class II mutations produce a highly proliferative disease with a block in differentiation, both hallmarks of AML.

The transcription factor CCAAT enhancer-binding protein alpha (CEBPA) is a master regulator of myeloid lineage commitment. CEBPA is recurrently mutated in AML and is a classic example of a Class II mutation[2]. The CEBPA gene comprises a single exon with an internal translational start site. Mutations in CEBPA cluster at the N- and C-terminus of the protein. N-terminal mutations typically result in a frameshift, leading to a premature stop codon and loss of expression of the long (p42) isoform of the protein but ongoing translation of the short isoform (p30). The p30 isoform lacks a crucial transactivation domain that represses cell cycle progression through a direct interaction with E2F[3]. In contrast, C-terminal mutations occur in the DNA-binding domain leading to loss of function and a blockade in granulocytic differentiation[2]. The most common pattern of mutation in AML is an N-terminal mutation on one allele and a C-terminal mutation on the other allele. This results in alterations in the ratio of the p42 to p30 CEBPA isoforms, changing the balance of differentiation and proliferation. AML with biallelic CEBPA mutations is associated with favorable prognosis, with approximately 50% of younger patients achieving a cure with chemotherapy alone. However, the precise determinants of relapse in this mutational context are unknown.

Recent studies have identified a high rate of co-occurrence of mutations in CEBPA with mutations the granulocyte-colony stimulating factor receptor (CSF3R)[4–7]. Approximately 20–30% of patients with CEBPA-mutant AML harbor a cooperating mutation in CSF3R. Although CSF3R mutations are often associated with biallelic CEBPA mutations, monoallelic cases also occur. Interestingly, monoallelic C-terminal CEBPA mutations are far more likely to co-occur with mutations in CSF3R. Patients with CSF3R/CEBPA mutant AML have inferior outcomes to those with mutant CEBPA alone, arguing that the presence of Class I mutations in CSF3R may be an important determinant of chemotherapy resistance and relapse[8]. The mutations in CSF3R most commonly occur in the membrane proximal region, and lead to ligand independent receptor dimerization and constitutive signaling via the JAK/STAT pathway. Similar to other Class I mutations, membrane proximal CSF3R mutations produce a myeloproliferative phenotype when present in isolation and are the major oncogenic driver of the disease chronic neutrophilic leukemia[9]. During normal myeloid development, G-CSF signaling via CSF3R leads to proliferation of myeloid precursors and neutrophilic differentiation. CEBPA is required for CSF3R-mediated transcription of myeloid specific genes, and myeloid differentiation arrests at the level of the common myeloid progenitor when CEBPA is deleted[10,11]. In spite of the established functional interdependence of CSF3R and CEBPA during normal hematopoiesis, the mechanism by which oncogenic mutations in these two genes interact to drive AML remains unknown.

Herein, we demonstrate that CSF3R and CEBPA cooperate to produce a highly proliferative immature myeloid leukemia in mice that phenocopies the human disease transcriptionally and morphologically. We establish that mutant CSF3R drives both proliferation and differentiation of maturing myeloid cells. In contrast, mutant CEBPA selectively blocks myeloid differentiation through inactivation of myeloid lineage enhancers. We further demonstrate that mutant CEBPA must occur prior to mutant CSF3R in order for AML to initiate. This confirms predictions based on clinical sequencing and provides a novel model to study oncogene order.

## Results

**Cooperativity Between CSF3R$^{T618I}$ and CEBPA Mutations.** We elected to study a representative N-terminal and C-terminal mutation (F82fs and V314VW), in combination with mutant CSF3R, as these specific mutations are known to co-occur in AML[5]. When expressed in mouse bone marrow cells via retroviral transduction, neither CEBPA$^{F82fs}$ nor CEBPA$^{V314VW}$ produced colonies in cytokine-free methylcellulose (Fig. 1a, b). As previously reported, CSF3R$^{T618I}$ produced a modest number of colonies in isolation[9]. The addition of CEBPA$^{V314VW}$, but not CEBPA$^{F82fs}$, dramatically augmented CSF3R$^{T618I}$-driven colony production and produced indefinite replating (Fig. 1c). These results were confirmed with a second C-terminal CEBPA mutation CEBPA$^{K313KR}$ (Fig. 1d). As CSF3R and CEBPA are both known to impact myeloid differentiation, we assessed cell morphology after 3 days of culture under cytokine-free conditions. Cells harboring empty vector and cells with CEBPA$^{F82fs}$ alone demonstrated neutrophilic morphology (Fig. 1e). CSF3R$^{T618I}$ increased the abundance of cells with larger morphology and immature granulocytic morphology. Expression of CEBPA$^{V314VW}$ produced a blast-like morphology, as did the combination of CSF3R$^{T618I}$ with either CEBPA mutation. Thus, it appears that mutant CSF3R drives proliferation of granulocytic precursors. Both N- and C-terminal mutations appeared to impact granulocytic differentiation in combination with CSF3R$^{T618I}$; however, CEBPA$^{V314VW}$ exerted a stronger effect, consistent with observed changes in colony production.

To determine how co-expression of both N- and C-terminal CEBPA mutations would alter the effects of CSF3R$^{T618I}$, we expressed all three mutations simultaneously. In this context, the addition of CEBPA$^{F82fs}$ mildly augmented CFU formation beyond that seen with CSF3R$^{T618I}$ and CEBPA$^{V314VW}$ (Fig. 1f, g). To establish whether the N- and C-terminal CEBPA mutations were producing the expected shifts in the CEBPA p30/p42 isoform ratio, we expressed each construct alone as well as the combination of isoforms in K562 cells. Western blotting for CEBPA in these cell lines revealed expression of the p30 isoform when CEBPA$^{F82fs}$ was present and expression of both the p42 and p30 isoforms when CEBPA$^{V314VW}$ was present (Supplementary Fig. 1A). We further validated that both mutant CSF3R and CEBPA were expressed in a cell line derived from mouse bone marrow colony-forming assay (Supplementary Fig. 1A). To determine whether the observed synergy between CSF3R$^{T618I}$ and CEBPA$^{V314VW}$ was more broadly generalizable, we looked for other mutations that co-occur with CEBPA in AML, which like CSF3R$^{T618I}$, also activate the JAK/STAT pathway. We identified a patient with CEBPA-mutant AML from a recently published data set that had a previously characterized activating mutation in JAK3 (JAK3$^{M511I}$)[12,13]. Simultaneous introduction of these mutations promoted cytokine-independent colony growth in vitro (Fig. 1h, i). Finally, we wanted to determine whether loss of CEBPA function is sufficient for enhanced oncogenesis in the context of mutant CSF3R. We therefore transduced bone marrow from CEBPA knockout mice with CSF3R$^{T618I}$, and found that they produced significantly more colonies than littermate control bone marrow (Fig. 1j, k).

To characterize gene expression changes driven by CSF3R$^{T618I}$ and CEBPA$^{V314VW}$, we performed RNA-seq on hematopoietic progenitors transduced with CSF3R$^{T618I}$, CEBPA$^{V314VW}$, or the combination of both. After 48 h of in vitro culture, cells expressing both oncogenes (marked by GFP and RFP) and not expressing mature lineage antigens (Lin−) were sorted by FACS prior to RNA

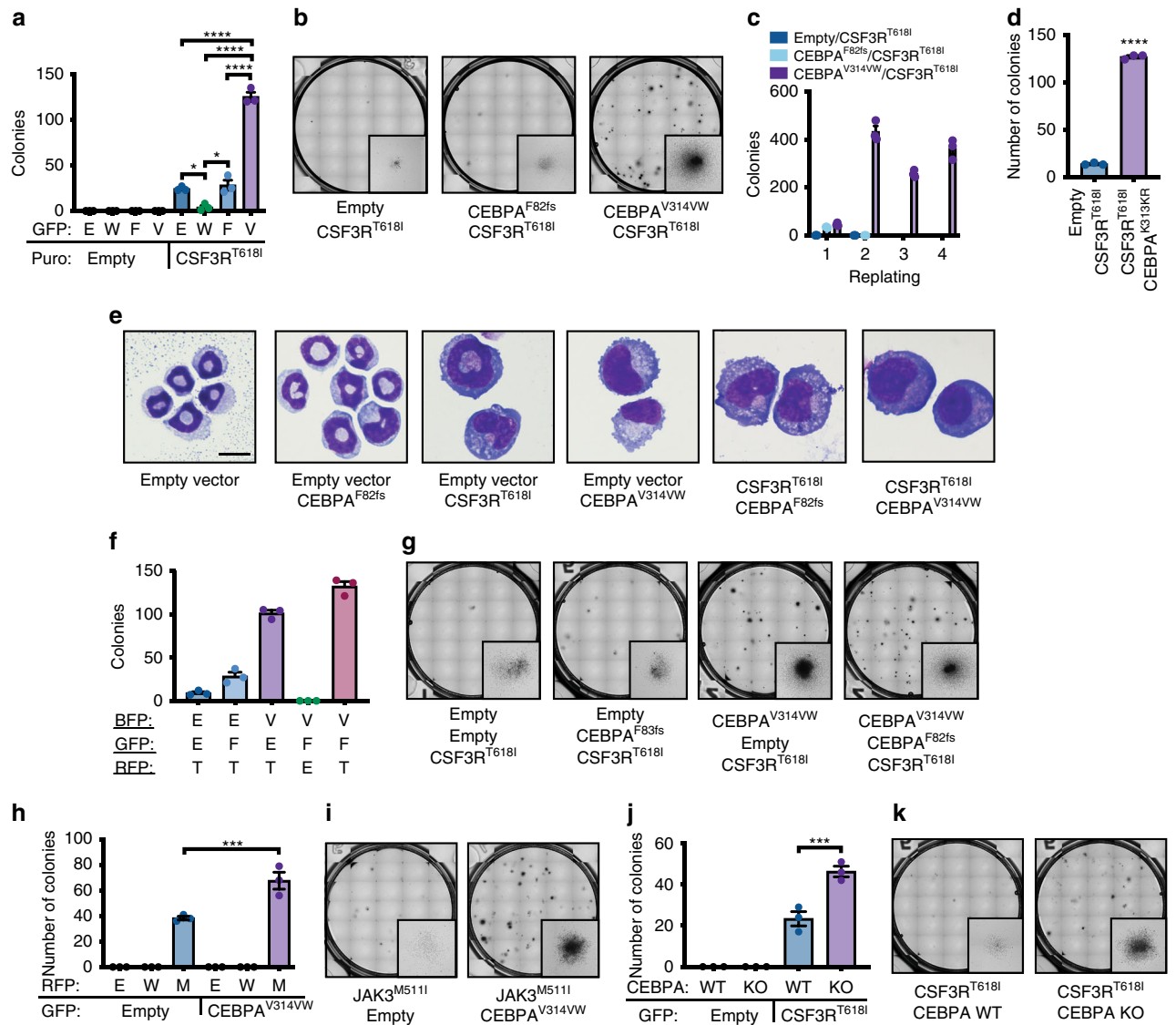

**Fig. 1** CSF3R$^{T618I}$ and CEBPA mutations cooperate in vitro. **a** Colony assay from mouse bone marrow transduced with CSF3R$^{T618I}$ and either CEBPA$^{WT}$ (W), CEBPA$^{F82fs}$ (F), or CEBPA$^{V314VW}$ (V) ($n = 3$/group). **b** Representative images for **a**. **c** Serial replating of colony assay in **a**. **d** Colony assay from mouse bone marrow transduced with CSF3R$^{T618I}$ and either empty vector or CEBPA$^{K313KR}$. **e** Example images from cytospins of conditions in **a** grown for 72 h in cytokine-free liquid culture (scale bar represents 10 μm). **f** Colony-forming assay from mouse bone marrow transduced with the combination of CSF3R$^{T618I}$ (T), CEBPA$^{F82fs}$ (F), CEBPA$^{V314VW}$ (V), two oncogene combinations and empty vector controls. In all, 1700 cells were plated per condition. **g** Representative images for **f**. **h** Colony assay from mouse bone marrow transduced with Empty Vector (E) JAK3$^{WT}$ (W), JAK3$^{M511I}$ (M), or CEBPA$^{V314VW}$. **i** Representative images of colony assay for **h**. **j** Colony assay from mouse bone marrow from CEBPA KO or Cre- littermate control bone marrow transduced with Empty Vector or CSF3R$^{T618I}$. **k** Representative colony assay images for **j**. In all cases $n = 3$/group and values are represented as mean with error bars representing SEM. Significance of comparisons assessed by ANOVA with Sidak post-test *$p < 0.05$, ***$p = 0.001$, ****$p < 0.0001$). Source data are provided as a Source Data file.

seq (Supplementary Fig. 1B, C). To describe the pattern of transcriptional changes driven by each oncogene, we used a linear model with an interaction term (Fig. 2a, Supplementary Data File 1), and identified 683 genes that were significantly up or down-regulated in response to CSF3R$^{T618I}$ ($q < 0.05$, Log$_2$ fold change >1 or <−1). We also identified 773 genes up- or down-regulated in response to CEBPA$^{V314VW}$ expression. Additionally, there were 570 genes that demonstrated an interaction between CSF3R$^{T618I}$ and CEBPA$^{V314VW}$ (effect less or more than additive). These interacting genes demonstrated a variety of patterns of regulation across all four oncogene conditions as demonstrated by K-means clustering (Fig. 2b). Of particular interest was a cluster of genes that were strongly up-regulated by CSF3R$^{T618I}$, but suppressed by co-expression of CEBPA$^{V314VW}$, as this mirrored the pattern of

myeloid differentiation. This cluster contained genes such as *Nos2*, *Hck*, *Stf1a*, and *Pla2g7*, all of which are expressed in mature neutrophils[14]. Also of interest was the gene *Myb*, a known driver of myeloid oncogenesis[15]. CEBPA$^{V314VW}$ increased the expression of *Myb* irrespective of the presence of CSF3R, suggesting that *Myb* plays a role in driving CEBPA-mediated oncogenesis. To confirm this, we overexpressed Myb in mouse bone marrow cells with and without CSF3R$^{T618I}$ (Supplementary Fig. 1D, E). While *Myb* did not drive cytokine-independent colony formation in isolation, it dramatically increased colony formation in combination with CSF3R$^{T618I}$, similar to that observed with CEBPA$^{V314VW}$.

To identify enriched transcriptional programs, we performed Gene Set Enrichment Analysis (GSEA) comparing CSF3R$^{T618I}$ to all other conditions (Fig. 2c, d, Supplementary Fig. 1F). CSF3R$^{T618I}$

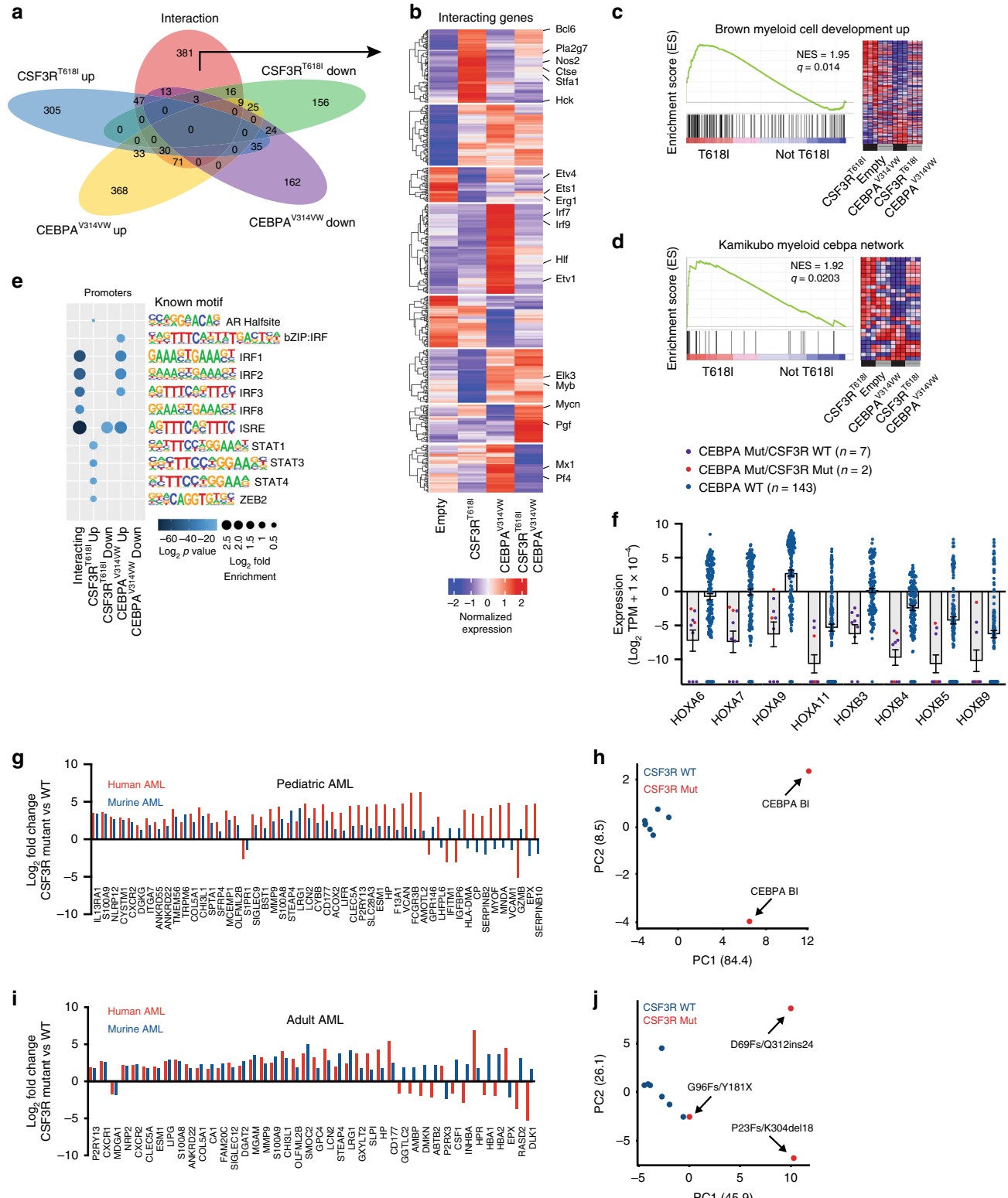

dramatically up-regulated genes associated with myeloid differentiation, while CEBPA$^{V314VW}$ strongly down-regulated them. In addition, genes associated with the wild-type CEBPA network followed a similar pattern (Fig. 2d). This suggests that myeloid differentiation in response to CSF3R$^{T618I}$ is, at least in part, dependent on CEBPA. We next performed motif enrichment analysis at the promoters of each category of differentially expressed genes (Fig. 2e). This identified a strong enrichment of STAT-binding sites in the promoters of CSF3R$^{T618I}$ up-regulated genes.

This is consistent with the known role of STAT3 as a crucial transcription factor downstream of oncogenic CSF3R mutations[9,16]. To confirm that STAT3 activation still occurs in the context of CSF3R$^{T618I}$ and CEBPA$^{V314VW}$ co-expression, we performed a western blot on bone marrow cells harboring both mutant CSF3R$^{T618I}$ and CEBPA$^{V314VW}$ growing in liquid culture after isolation from colony assay. This confirmed an increase in phosphorylated STAT3 relative to normal bone marrow (Supplementary Fig. 1G). Given the results from the GSEA, we were

**Fig. 2** Mutant CEBPA blocks myeloid differentiation in response to CSF3R. **a** Venn diagram of differentially expressed genes from RNA-seq on lineage-negative mouse bone marrow transduced with empty vector, CSF3R[T618I], CEBPA[V314VW] or the combination of oncogenes. **b** Hierarchical clustering of interacting genes. **c, d** Select enriched gene sets in CSF3R[T618I] vs. other categories NES = normalized enrichment score ($q < 0.05$). GSEA $p$ value calculated by empirical permutation test and FDR adjusted. **e** Motif enrichment at promoters of differentially expressed (DE) genes. Top five motifs per category with $q < 0.05$ shown. $P$ values generated via comparison to binomal distribution and FDR adjusted. **f** Expression of differentially expressed HOX genes in pediatric AML patients harboring CEBPA mutations. **g** Expression of genes differentially expressed in both murine and pediatric human CSF3R/CEBPA AML as compared with CEBPA-mutant CSF3R-WT AML. **h** PCA analysis of pediatric CEBPA mutant AML using convergent human-mouse gene set. Cases with biallelic CEBPA mutations are indicated by CEBPA-Bi. **i** Expression of genes differentially expressed in both murine and adult human CSF3R/CEBPA AML as compared with CEBPA-mutant CSF3R-WT AML. **j** PCA analysis of adult CEBPA mutant AML using convergent human–mouse gene set. Specific CEBPA mutations are indicated by the text. Source data are provided as a Source Data file.

somewhat surprised that CEBPA motifs were not detected at the promoters of any of the groups of DE genes, suggesting that CEBPA[V314VW] may impact CSF3R[T618I]-induced myeloid differentiation through binding to non-promoter regulatory regions. To determine, whether CEBPA binding was significantly associated with any of the categories of differentially expressed genes, we performed a permutation analysis using published ChIP-seq data from mouse granulocyte–macrophage progenitors (GMPs)[17]. This analysis revealed that the interacting genes, CSF3R[T618I] up genes, and CEBPA[V314VW] down genes were all found in closer proximity to CEBPA peaks than would be expected by random chance (Supplementary Fig. 1H).

To validate these gene findings in human AML, we examined RNA-seq data from the pediatric TARGET initiative[18]. A total of 152 patients were evaluated, including 7 patients with CEBPA mutations and 2 patients with both CSF3R and CEBPA mutations. Given the small number of patients, both monoallelic and biallelic CEBPA mutations were considered together. Differential gene expression revealed markedly decreased *HOX* gene expression in CEBPA mutant samples compared with CEBPA WT samples, a well-established finding in adult CEBPA mutant AML (Fig. 2f, Supplementary Data File 2)[19]. Comparison of CEBPA mutant/ CSF3R WT and CEBPA mutant/CSF3R mutant patient samples revealed 913 differentially expressed genes (Supplementary Data File 2). Comparison of these differentially expressed genes to those identified in mouse revealed 52 ortholog pairs with differential expression driven by mutant CSF3R (Fig. 2g). Of these, 75% demonstrated concordant regulation between mouse and human. A similar analysis was performed on adult CEBPA/CSF3R mutant AML samples from the Leucegene cohort, which also demonstrated that the preponderance of orthologous gene pairs display concordant regulation (68%; Fig. 2i)[4]. Both human/mouse concordant gene sets were able to independently segregate CSF3R mutant from wild-type samples by PCA (Fig. 2h, j). As CEBPA is known to regulate CSF3R expression, we examined CSF3R expression in CEBPA WT and Mutant AML samples (Supplementary Fig. 1I)[20]. Both subsets of patients robustly expressed CSF3R and no significant difference in expression was observed on the basis of CEBPA mutational status. Thus, although wild-type CEBPA does regulate the activity of the CSF3R promoter, it appears that CSF3R expression is not dramatically altered in AML by the presence of putative loss-of-function mutations in CEBPA.

**JAK/STAT activation and CEBPA dysregulation drive AML.** To establish whether oncogenic CSF3R and CEBPA mutations collaborate in vivo to produce AML, we performed murine bone marrow transplantation with fetal liver hematopoietic cells harboring compound heterozygous CEBPA mutations (CEBPA[K/L]) or wild-type CEBPA retrovirally transduced with CSF3R[T618I][21]. As previously reported, transplantation with CEBPA[K/L] cells alone leads to disease onset with a median latency of

approximately 1-year post transplant (Fig. 3a). In contrast, mice transplanted with CEBPA[K/L] cells harboring mutant CSF3R developed AML that was uniformly lethal by 1 year and occurred far more rapidly in the majority of cases. When leukemia developed, it was accompanied by leukocytosis, splenomegaly and morphologic myeloblasts in the bone marrow (Fig. 3b–d).

As CSF3R mutations appear to correlate most strongly with the presence of C-terminal CEBPA mutations in human AML, and our colony assay data presented in Fig. 1 demonstrate more potent oncogenic activity in vitro we investigated whether similar biology occurs in in vivo. We therefore also performed retroviral bone marrow transplantation with retrovirally introduced CSF3R[T618I], CEBPA[F82fs], CEBPA[V314VW], and JAK3[M511I] with corresponding single oncogene and empty vector controls. Mice receiving cells transduced with CSF3R[T618I] and CEBPA[V314VW] developed a myeloid leukemia that was uniformly lethal by day 14 post-transplant and was associated with leukocytosis and marked splenomegaly (Fig. 3e, f, Supplementary Fig. 2A). In the bone marrow, normal hematopoiesis was completely replaced by large cells with myeloblastic morphology (Fig. 3g, Supplementary Figs. 2B and 3A–C). Mice harboring CSF3R[T618I] alone or in combination with CEBPA[F82fs] developed disease with a long latency, associated with leukocytosis, variable splenomegaly, mature immunophenotype, and morphologic neutrophils in the peripheral blood and bone marrow (Fig. 3e, g). CEBPA[V314VW] alone resulted in a long-latency myeloid leukemia with immature histologic features at the time of ultimate disease development (Fig. 3e, g). Similar to CSF3R[T618I], JAK3[M511I] in combination with CEBPA[V314VW] also produced a rapidly lethal myeloid leukemia with markedly reduced latency compared with expression of either mutation alone, again with the accumulation of marrow myeloblasts (Fig. 3e–g).

CSF3R mutations are associated with chronic neutrophilic leukemia (defined by an abundance of mature neutrophils) yet also accelerate AML formation in the context of mutant CEBPA. Therefore, we were interested in understanding how the presence of mutant CEBPA changes the oncogenic program of mutant CSF3R. Unfortunately, the AML that develops from CSF3R[T618I]/CEBPA[V314VW] co-expression results in a uniformly lethal AML when control mice are still pancytopenic. Comparison with untransplanted control mice reveals that CSF3R/CEBPA mutant AML cells are CD11b positive and GR-1 dim and completely replace the normal hematopoietic hierarchy (Supplementary Fig. 3A–C). However, to make a direct comparison, we transplanted syngeneic recipient mice with either 100,000 cells expressing CSF3R[T618I] alone or 10,000 cells expressing CSF3R[T618I] and CEBPA[V314VW] (Fig. 4a). This log-fold reduction in cell dose delayed the timing of disease onset to approximately 20 days, allowing for side by side comparison of these two groups. While CSF3R[T618I] alone mice demonstrated an abundance of mature neutrophils (marked by high levels of CD11b and GR-1), the leukemic blasts seen in CSF3R[T618I]/CEBPA[V314VW] mice exhibited lower levels of GR-1 staining consistent with an immature myeloid

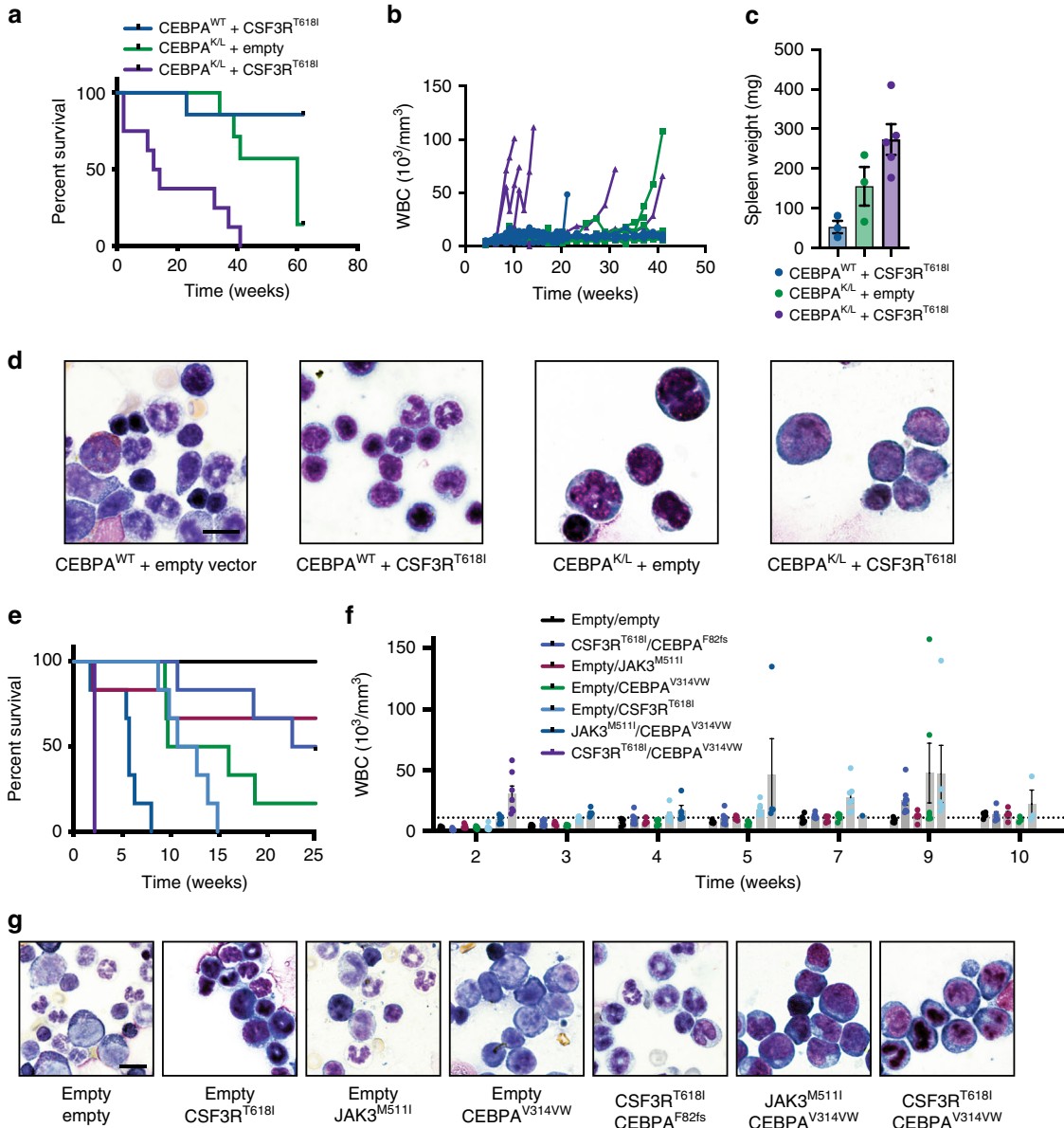

**Fig. 3** JAK/STAT activation and CEBPA mutations cooperate in vivo. **a** Survival of mice transplanted with CEBPA wild type or CEBPA$^{K/L}$ fetal liver cells transduced with empty vector or CSF3R$^{T618I}$ ($n = 7$–8/group). **b** WBC counts from mice in A. **c** Spleen weights from moribund mice in **a** ($n = 3$–5/group). **d** Example bone marrow smears for mice in **a** (scale bar represents 10 μm). **e** Survival of mice transplanted with 100,000 cells containing Empty Vector, JAK3$^{M511I}$ + Empty Vector, CEBPA$^{V314VW}$ + Empty Vector, JAK3$^{M511I}$ + CEBPA$^{V314VW}$, CSF3R$^{T618I}$ + Empty vector, CSF3R$^{T618I}$ + CEBPA$^{F82fs}$, or CSF3R$^{T618I}$ + CEBPA$^{V314VW}$ ($n = 5$–6/group). **f** WBC counts of mice harboring mutation combinations in **e** ($n = 5$–6/group at experiment start). **g** Example bone marrow smears from mice in **e** (scale bar represents 10 μm). Values in bar graphs are represented as mean with error bars representing SEM. Source data are provided as a Source Data file.

phenotype (Fig. 4b–d, Supplementary Fig. 3D). Marked spleno-megaly was observed in mice transplanted with CSF3R$^{T618I}$/CEBPA$^{V314VW}$ cells but not in mice receiving CSF3R$^{T618I}$ alone cells (Fig. 4e, f). To establish whether the leukemic blasts generated by co-expression of CSF3R$^{T618I}$ and CEBPA$^{V314VW}$ were capable of indefinite self-renewal, we performed serial transplantation studies. These experiments revealed that the leukemia was capable of disease initiation in up to quaternary recipients (Supplementary Fig. 3E–G).

**CEBPA mutations disrupt myeloid lineage enhancers**. During normal hematopoietic development, CEBPA is responsible for establishing the enhancer landscape that permits myeloid differentiation[22]. We therefore hypothesized that CEBPA$^{V314VW}$

blocks differentiation by inhibiting priming or activation of myeloid lineage enhancers. To test this, we utilized murine HoxB8-ER cells, which mimic GMPs and differentiate down the neutrophilic lineage to become CD11b and GR-1 positive after estrogen withdrawal[23]. In this model, expression of CSF3R$^{T618I}$ accelerated differentiation, while CEBPA$^{V314VW}$ blocked differentiation, as measured by CD11b and GR-1 expression (Supplementary Fig. 4A, B). The combination of mutations produced maturation arrest, similar to CSF3R/CEBPA mutant murine leukemia. We confirmed that neither oncogene was changing the expression of the other by qPCR (Supplementary Fig. 4C), consistent with our findings in human CEBPA mutant AML. Thus, HoxB8-ER cells expressing CSF3R$^{T618I}$ and CEBPA$^{V314VW}$ recapitulate the phenotype seen in our in vivo model.

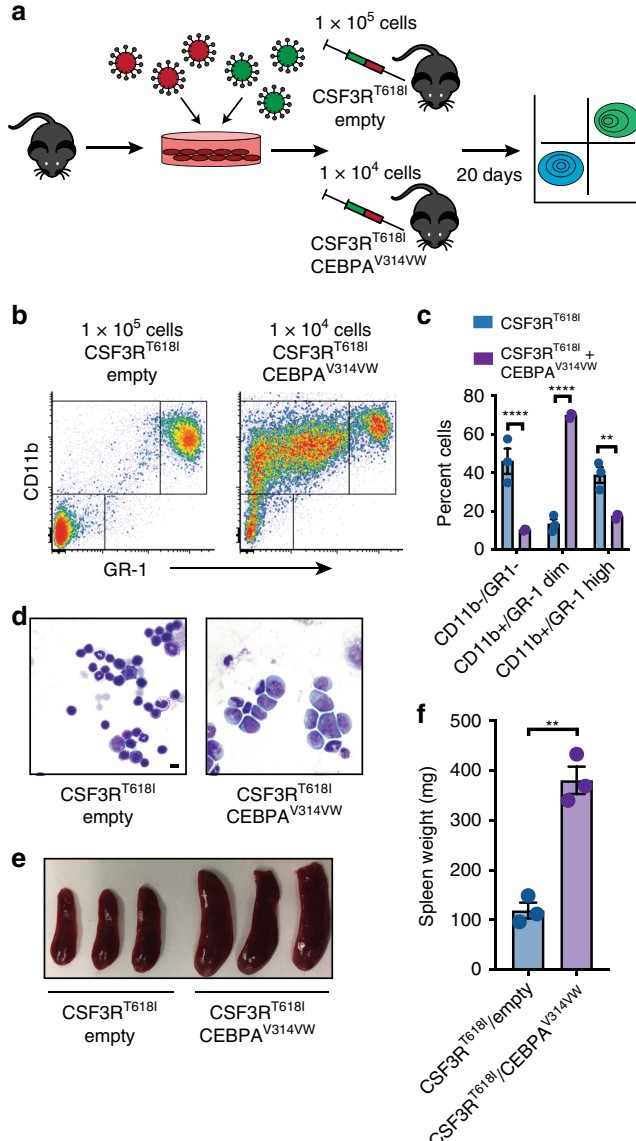

**Fig. 4** CEBPA mutations block differentiation in response to mutant CSF3R. **a** Experimental strategy to directly compare the differentiation of cells harboring CSF3R$^{T618I}$ and CSF3R$^{T618I}$/CEBPA$^{V314VW}$. **b** Bone marrow flow cytometry comparing myeloid differentiation markers in mice transplanted under conditions in **a**. **c** Quantification of populations in **b**. **d** Example bone marrow histology from mice transplanted in **a** (scale bar represents 10 μm). **e** Spleens from mice in **a**. **f** Spleen weights from mice in **a**. Values are represented as mean with error bars representing SEM. Significance of comparisons assessed by Students T-test or ANOVA with Sidak post-test as appropriate **$p < 0.01$, ****$p < 0.0001$. Source data are provided as a Source Data file.

To study enhancer dynamics in the context of these mutations, we performed chromatin immunoprecipitation sequencing (ChIP-seq) for H3K4me1, H3K4me3, and H3K27ac in HoxB8-ER cells transduced with CSF3R$^{T618I}$, CEBPA$^{V314VW}$, both mutations, or empty vectors. We identified 42,124 active enhancers (defined as high H3K4me1, low H3K4me3, and high H3K27ac) across all four conditions (Supplementary Data File 3). To validate these findings in human AML, we mapped our enhancers to orthologous human genomic coordinates and assessed overlap of enhancers only active in the CSF3R$^{T618I}$/CEBPA$^{V314VW}$ condition with those present in human AML

with mutant CEBPA[24]. We observed a significant enrichment of overlap between our mouse AML enhancers with those found in human CEBPA mutant AML, suggesting conserved biology (Supplementary Fig. 5A). Furthermore, our murine leukemic enhancers demonstrated less overlap than expected through random chance with those found in human CEBPA wild-type AML.

We next focused on enhancers that were active exclusively in one condition (Condition-Specific Enhancers, Fig. 5a, b, Supplementary Fig. 5B). Gene ontology analysis on the 2237 CSF3R$^{T618I}$-specific enhancers revealed enrichment for terms associated with immune responses and phagocytosis, demonstrating that these genes are associated with the mature neutrophil phenotype (Fig. 5c, Supplementary Data File 4). To understand whether CEBPA$^{V314VW}$ blocks myeloid differentiation through disruption of CSF3R$^{T618I}$-specific enhancers and associated gene expression, we performed microarray gene expression analysis on HoxB8-ER cells harboring each oncogene in isolation or the combination, and the expression of key genes was validated by qPCR (Supplementary Data File 5, Supplementary Fig. 5C). Genes associated with mature myeloid phenotypes, such as Nos2, Hck, Bcl6, and Pla2g7, displayed increased expression in the CSF3R$^{T618I}$ condition, and were repressed with co-expression of CEBPA$^{V314VW}$. To globally assess whether CSF3R target genes were associated with CSF3R$^{T618I}$-specific enhancers, we performed hierarchical clustering on genes that were differentially expressed in one or more treatment condition. We then examined enrichment of each condition-specific enhancer group across these clusters (Supplementary Fig. 5D, Supplementary Data File 5). This analysis demonstrated globally that activated enhancers are associated with increased expression of the nearest gene.

To identify possible regulators of enhancer activation, we performed transcription factor motif enrichment. This analysis demonstrated enrichment of CEBPA motifs in CSF3R$^{T618I}$-specific enhancers, but not in any other group (Fig. 5d). This suggests that CEBPA$^{V314VW}$ blocks differentiation by disrupting the interaction of wild-type CEBPA with CSF3R$^{T618I}$-specific enhancers. To provide further evidence for this hypothesis, we utilized published CEBPA ChIP-seq data from GMPs (the nearest normal cell to HoxB8 immortalized progenitors)[17]. We found strong enrichment of CEBPA peaks overlapping CSF3R$^{T618I}$-specific enhancers, but not the other condition-specific enhancer groups (Fig. 5e, f). These data suggest that these differentiation-associated enhancers are regulated by direct CEBPA binding during normal hematopoiesis and the presence of mutant CEBPA prevents activation of these enhancers.

Our data suggest that differentiation in response to oncogenic CSF3R signaling requires the action of wild-type CEBPA at differentiation-associated enhancers. However, the proliferative effects of CSF3R$^{T618I}$ are likely independent of CEBPA. To identify the effectors of CSF3R-induced proliferation, we examined enhancers activated by CSF3R$^{T618I}$ irrespective of the presence of mutant CEBPA. Gene ontology analysis on this subset of enhancers identified multiple pathways associated with cell cycle progression (Fig. 5g). One likely driver of this CSF3R-induced proliferation is the transcription factor E2f2 which is strongly induced by CSF3R$^{T618I}$ in the presence and absence of CEBPA$^{V314VW}$ (Fig. 5h). Correlation with published STAT3 ChIP seq data from murine myeloid cells reveals STAT3 binding to an enhancer at the E2f2 locus that is activated by CSF3R$^{T618I}$ (Fig. 5h)[25]. Thus, it seems likely that key components of the proliferative program downstream of CSF3R$^{T618I}$ are driven via activation of enhancers associated with cell cycle progression, possibly in a STAT3-dependent manner.

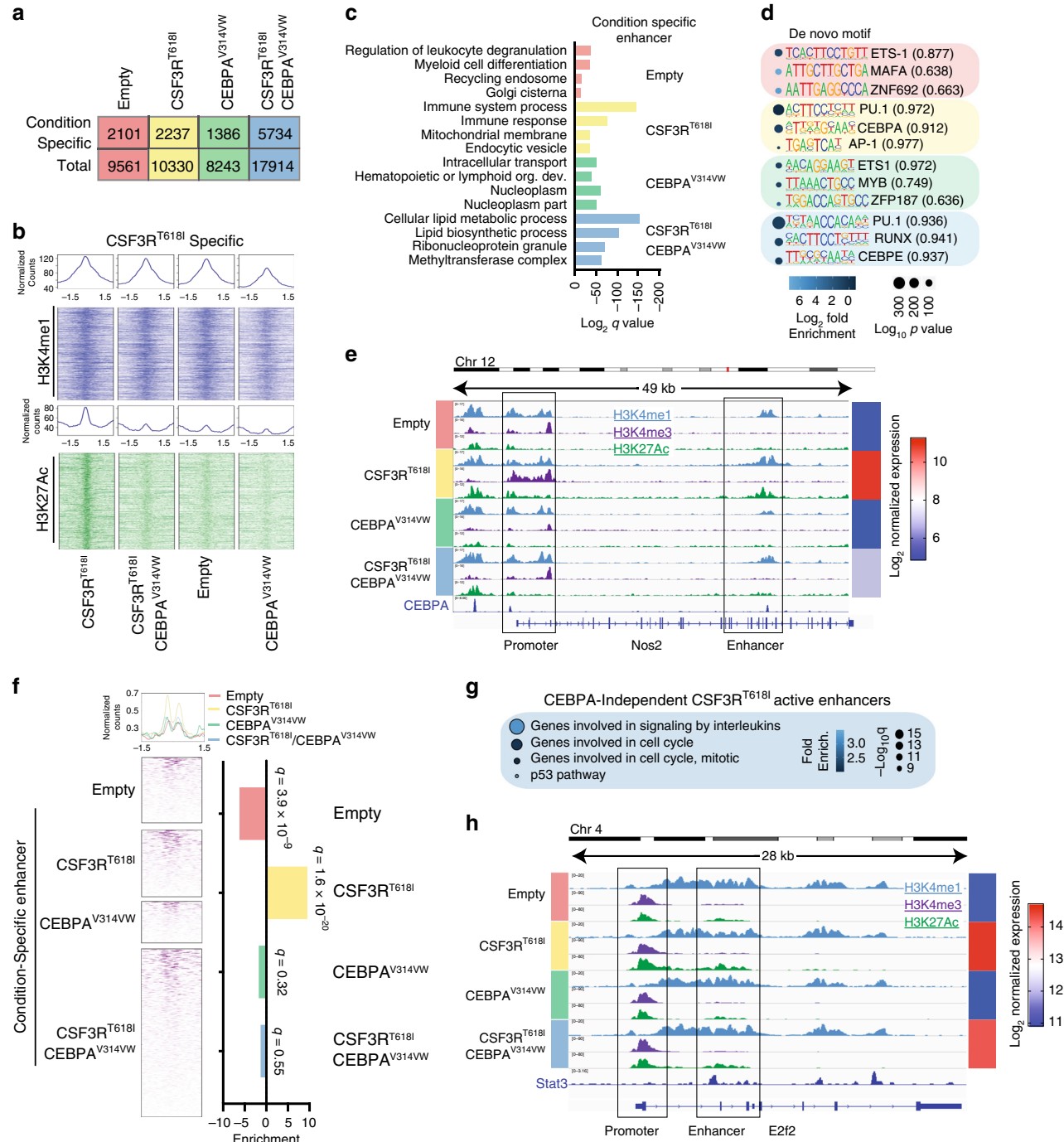

**Fig. 5** CEBPA mutations disrupt activation of myeloid lineage enhancers. **a** Number of condition-specific enhancers identified in this study. **b** ChIP-seq heat maps for H3K4me1 and H3K27ac at +1.5 kb around the center of CSF3R[T618I]-specific enhancers, across treatment conditions. ChIP tracks display fold enrichment relative to corresponding input. **c** Gene ontology analysis for condition-specific enhancers. *P* values generated via comparison to binomal distribution and FDR adjusted. **d** De novo motif enrichment in condition-specific enhancers. Values in parenthesis correspond to motif match score. *P* values generated via comparison to binomal distribution and FDR adjusted. **e** The epigenetic landscape at the Nos2 locus demonstrates condition-specific promoter and enhancer activation. **f** Assessment of CEBPA ChIP-seq peak overlap with condition specific enhancers by $\chi^2$ test with Pearson residual values plotted. Adjusted *p* values (by the method of Holm–Bonferroni) are displayed above bars. **g** Gene ontology analysis for CEBPA-independent CSF3R[T618I] active enhancers. **h** The epigenetic landscape at the E2f2 locus demonstrates enhancer activation in response to CSF3R[T618I]. Source data are provided as a Source Data file.

**CEBPA mutations must precede mutations in CSF3R**. Enhancer priming and activation precede promoter activation, suggesting that epigenetic changes at enhancers must occur as early events[26]. Clinical sequencing data demonstrate that CEBPA mutations frequently occur at higher variant allele frequencies than CSF3R mutations[4,5]. This has led to the prediction that CEBPA mutations occur early in disease development. Conversely, however, it is possible that CEBPA mutations could be acquired late during blast-crisis transformation of chronic neutrophilic leukemia. To evaluate the impact of mutation order on

AML initiation, we developed a Cre-inducible retroviral vector. We paired this vector with hematopoietic cells from Rosa 26$^{ERT2}$-Cre mice, in which recombination is activated via tamoxifen administration (Fig. 6a). We co-expressed this inducible vector with a constitutive vector, enabling the study of two distinct mutational orders of acquisition: CSF3R$^{T618I}$-first and CEBPA$^{V314VW}$-first. We first assessed the impact of mutation order on bone marrow colony formation by plating both ordered combinations in the presence of 4-hydroxytamoxifen (4-OHT), thus delaying the expression of the second oncogene by approximately 48 h. Strikingly, we found that CSF3R$^{T618I}$-first produced far fewer colonies than CEBPA$^{V314VW}$-first (Fig. 6b, c). Numerically and morphologically, CSF3R$^{T618I}$-first colonies were closer to CSF3R$^{T618I}$-only. This finding was not due to differential expression of CEBPA from the inducible and constitutive vectors, as both showed equivalent CEBPA expression (Fig. 6d).

To establish the transcriptional profile of these two distinct mutation orders, we performed RNA-seq on lineage negative mouse bone marrow transduced with CSF3R$^{T618I}$-only, CSF3R$^{T618I}$-first, and CEBPA$^{V314VW}$-first, cultured for 48 h with 4-OHT (Supplementary Fig. 6A, B, Supplementary Data File 6). Unsupervised clustering by Euclidean distance revealed that CSF3R$^{T618I}$-only and CSF3R$^{T618I}$-first cluster together and are globally distinct from CEBPA$^{V314VW}$-first (Fig. 6e). The expression of differentiation-associated genes (*Nos2*, *Hck*, *Bcl6*, *Pla2g7*) in CSF3R$^{T618I}$-first cells was intermediary between CSF3R$^{T618I}$-only and CEBPA$^{V314VW}$-first (Fig. 6f, Supplementary Data File 6). GSEA revealed enrichment for numerous signatures associated with myeloid differentiation and inflammatory responses in CSF3R$^{T618I}$-first cells (Fig. 6g, Supplementary Fig. 6C). CEBPA$^{V314VW}$-first cells demonstrated enrichment for signatures associated with cell cycle progression and stem/progenitor cell phenotype. Taken together, these studies demonstrate a profound effect of mutation order on gene expression and hematopoietic phenotype in vitro.

We next explored the contribution of mutation order to disease latency and phenotype in vivo. Mice were transplanted with CSF3R$^{T618I}$-first or CEBPA$^{V314VW}$-first cells and allowed to recover for 4-weeks post-transplant. After 4-weeks, the second oncogene was induced with tamoxifen (Fig. 7a). CEBPA$^{V314VW}$-first mice succumbed to lethal myeloid leukemia with a median survival of 3.5 weeks post-tamoxifen (Fig. 7b). CEBPA$^{V314VW}$-first leukemia was associated with bone marrow and peripheral blood blasts with an immature immunophenotype as well as splenomegaly (Fig. 7c–g, Supplementary Fig. 7A, B). In contrast, CSF3R$^{T618I}$-first mice displayed a differentiated myeloid immunophenotype when sacrificed at an early time point. Only one CSF3R$^{T618I}$-first mouse developed leukemia at 7 weeks post-tamoxifen administration. Bone marrow cytology revealed a blast-like morphology; however, flow cytometry revealed increased GR-1 staining, indicative of a higher degree of myeloid maturation (Fig. 7c, e). Collectively, these data reveal that when CEBPA mutations are introduced after mutations in CSF3R, they are unable to fully block myeloid differentiation. Importantly, this impaired ability to block differentiation disrupts the development of AML in vivo.

## Discussion

Our study adds to a growing body of data demonstrating that enhancer biology is integral to the development of hematologic malignancies. While AML is very heterogeneous from a genomic standpoint, recent work demonstrates that there are only a few epigenetic disease subtypes[24,27]. Although multiple global epigenetic regulators are recurrently mutated in AML, these have little impact on the organization of the epigenome[24,27]. Instead,

mutations in lineage determining transcription factors are a major determinant of clustering. Thus, understanding the epigenetic dysfunction associated with transcription factor mutations in AML may provide broad insight into therapeutic approaches.

N- and C-terminal CEBPA mutations exert distinct biological roles in leukemia initiation[21,28]. While it is clear that reduced CEBPA activity is potently oncogenic, CEBPA knockout mice fail to develop AML[29]. Similarly, mice with homozygous C-terminal mutations (putatively loss of function) develop leukemia with an exceedingly long latency[21]. Interestingly, the phenotype of these leukemias is erythroid rather than myeloid. This is consistent with data from MLL-rearranged leukemias, where CEBPA is requisite for entrance into the myeloid lineage and leukemia initiation[30,31]. As the majority of patients with CEBPA mutant AML harbor combined N- and C- terminal CEBPA mutations, it is possible that the N-terminal CEBPA mutation provides sufficient residual myeloid differentiation potential to enter the myeloid lineage and initiate AML. This relationship becomes even more complex when CSF3R mutations are considered. While the majority of CSF3R mutations occur in CEBPA-bi cases there are a number of CSF3R mutant CEBPA C-terminal monoallelic cases as well[5]. Our data demonstrate that CSF3R$^{T618I}$ potently synergizes with biallelic CEBPA mutations to induce AML. Our retroviral model best represents monoallelic cases (with the caveats of ectopic expression), and demonstrates that C- but not N-terminal CEBPA mutations exhibit synergy with mutant CEBPA confirming the high degree of clinical association between these mutations. It is possible that mutant CSF3R provides both a myeloid commitment signal and proliferative advantage, thus rendering N-terminal CEBPA mutations less crucial for AML initiation. Going forward, it will be crucial to define the biological interaction with mutant CSF3R and N- and C-terminal mutations present in the endogenous locus to understand whether these mutant forms of CEBPA demonstrate differential binding or recruitment of cofactors to critical differentiation-associated enhancers.

Although we focused on differentiation-associated enhancers driven by the CSF3R$^{T618I}$, our study revealed a second set of enhancers that were activated exclusively in the presence of both mutant CSF3R and CEBPA. Both subsets of enhancers demonstrated strong enrichment of PU.1 motifs, consistent with the known role of this transcription factor in driving myeloid development. In normal hematopoiesis, PU.1 and CEBPA cooperate to open myeloid lineage enhancers with PU.1 performing pioneering function in early progenitors and CEBPA assuming this role late[22]. Thus, it is likely that certain early myeloid lineage enhancers can be activated by CSF3R$^{T618I}$ in a CEBPA-independent manner via PU.1. Another interesting finding was the enrichment of RUNX motifs in enhancers activated only in the presence of both oncogenes. The RUNX family of transcription factors are critical to normal hematopoietic development, and in addition to being the targets of chromosomal translocations in core-binding factor AML, also frequently harbor point mutations in AML[32]. Interestingly, Runx1 haploinsufficiency leads to hypersensitivity to G-CSF, suggesting a negative feedback role[33]. The role of RUNX transcription factors in driving CEBPA/CSF3R mutant AML is an interesting area for future work.

We present the first direct evidence that the order in which oncogenic mutations occur is a major determinant of leukemia development (Fig. 8). Our finding that myeloid differentiation blockade can only occur with a distinct mutational order may also be a broadly conserved mechanism that applies to Class I and Class II mutation pairings. As nearly all prior studies have investigated mutation cooperation in the setting of simultaneous introduction, it is likely that important aspects of

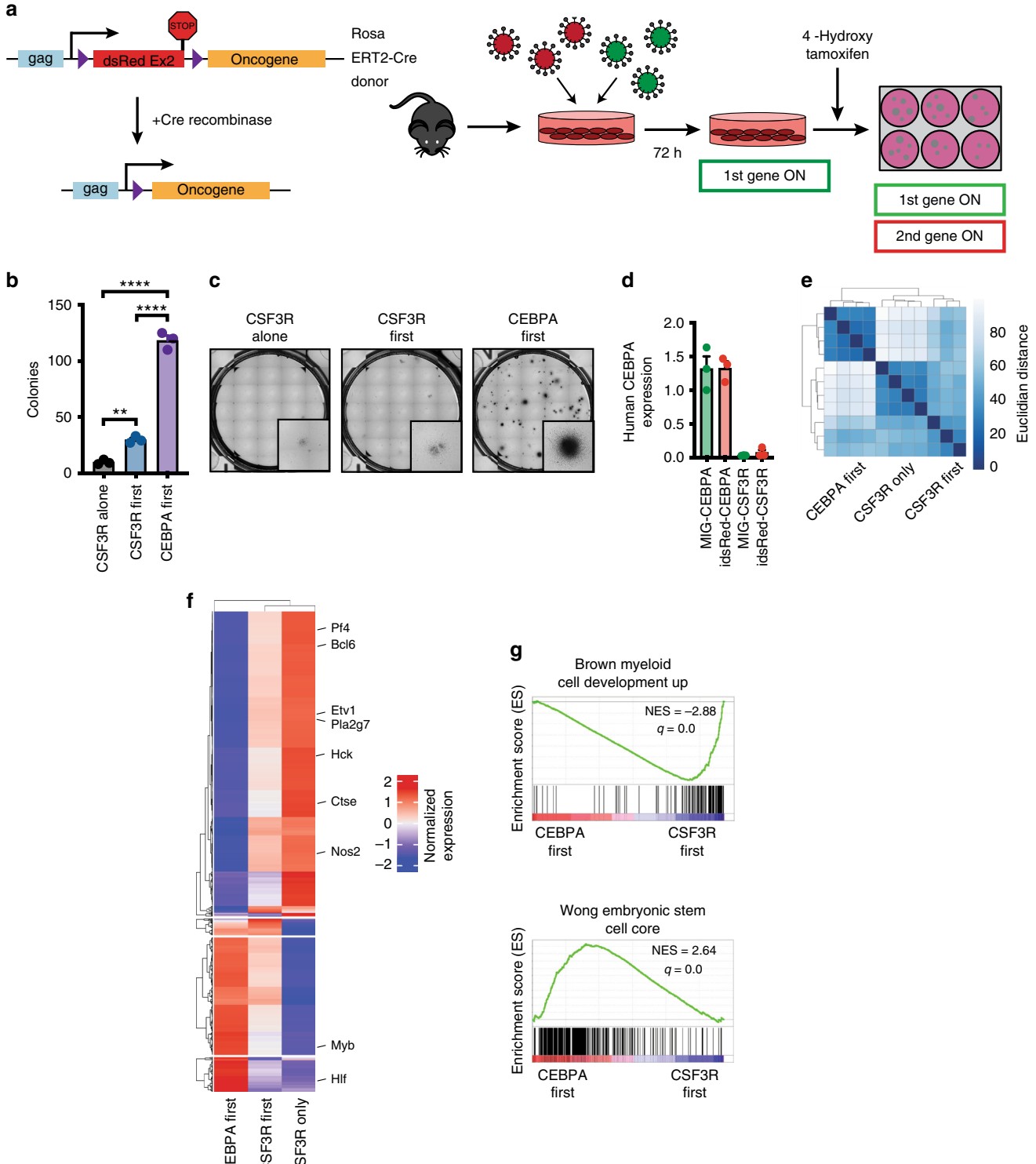

**Fig. 6** CEBPA mutations must precede mutations in CSF3R to block differentiation. **a** Diagram of order of acquisition system. Oncogenes in the MIG (GFP+) vector tagged with GFP are constitutively expressed, and oncogenes in the idsRed vector are expressed only after Cre-mediated recombination. Expression of the second oncogene can be induced by culture in 4-OHT. **b** Colony assay from mouse bone marrow transduced with CEBPA First (MIG-CEBPA$^{V314VW}$ + idsRed-CSF3R$^{T618I}$) CSF3R First (MIG-CSF3R$^{T618I}$ + idsRed-CEBPA$^{V314VW}$) and CSF3R only (MIG-CSF3R$^{T618I}$ + idsRed Empty) and plated in 4-OHT ($n = 3$/group). **c** Representative images of colony assay. **d** Human CEBPA and CSF3R expression in mouse bone marrow expressing MIG-CEBPA$^{V314VW}$, idsRed-CEBPA$^{V314VW}$, MIG-CSF3R$^{T618I}$ or idsRed-CSF3R$^{T618I}$ measured by TaqMan quantitative PCR ($n = 3$/group). **e** Clustering by Euclidian distance for RNA sequencing performed on lineage negative mouse bone marrow expressing CEBPA-first, CSF3R-first, or CSF3R-only ($n = 3$–4/group). **f** K-means clustering of top 750 differentially expressed genes per pairwise comparison ($q < 0.05$, log$_2$Fold Change < −1 or >1) ($n = 3$–4/group). **g** Representative enriched gene sets from GSEA performed on samples from **e** ($n = 3$–4/group). GSEA $p$ value calculated by empirical permutation test and FDR adjusted. In all cases, values are represented as mean with error bars representing SEM. \*\*$p < 0.01$, \*\*\*\*$p < 0.0001$, as measured by ANOVA with Sidak's post-test. Source data are provided as a Source Data file.

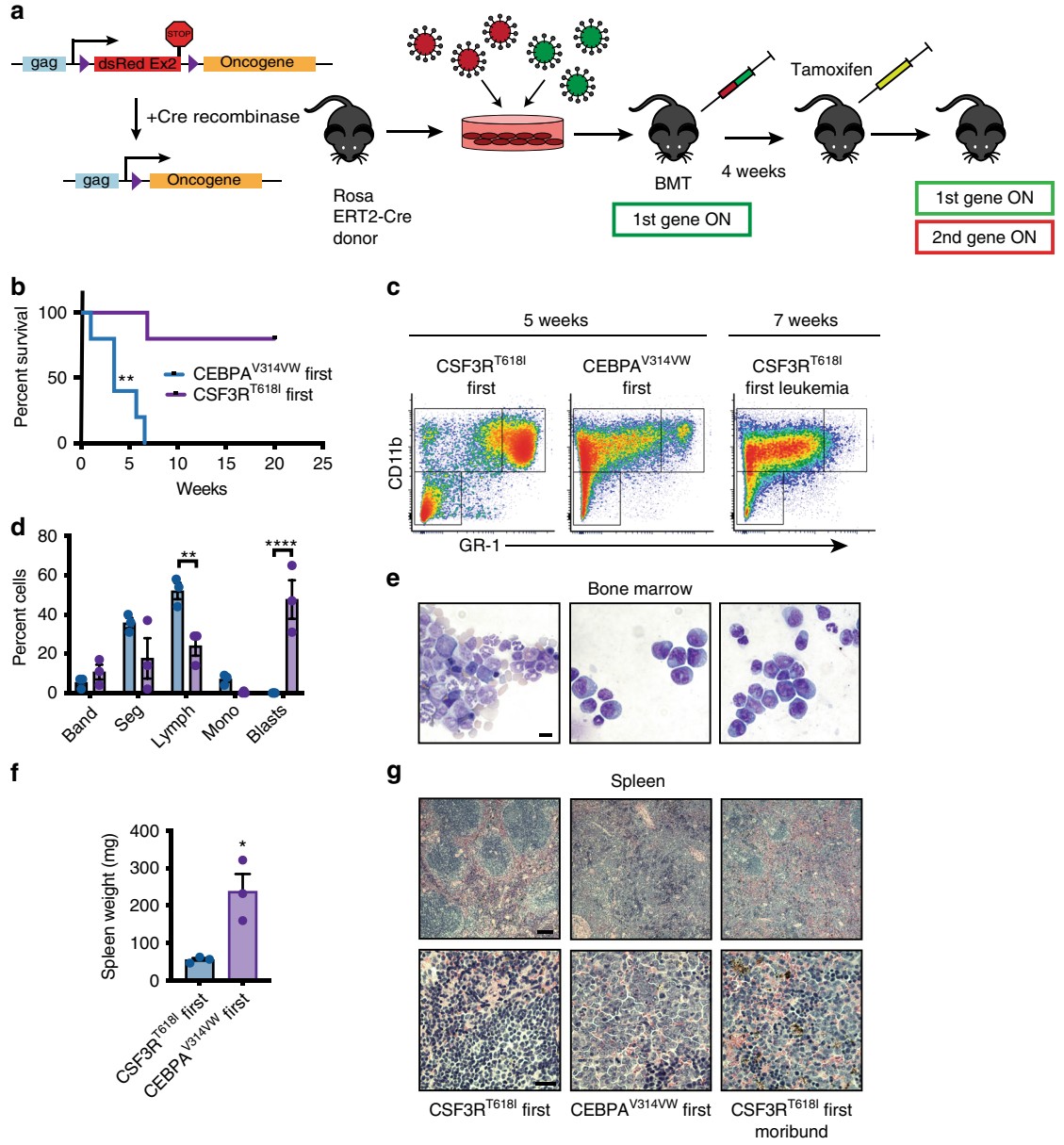

**Fig. 7** CEBPA mutations must precede mutations in CSF3R for AML initiation. **a** Diagram of order of acquisition system. Oncogenes in the MIG vector tagged with GFP are constitutively expressed and oncogenes in the idsRed vector are expressed only after Cre mediated recombination. Expression of the second oncogene can be induced by administration of tamoxifen. Mice were administered 75,000 GFP/RFP-positive cells. **b** Survival reported as time from day 1 of tamoxifen induction ($n = 5$/group). Statistical significance calculated by Log Rank test, $**p < 0.01$. **c** Expression of CD11b and GR-1 in bone marrow at 5 weeks in moribund CEBPA$^{V314VW}$-first mice, healthy CSF3R$^{T618I}$-first mice or at 7-weeks in leukemic CSF3R$^{T618I}$-first mice (only one animal represented by CSF3R-first leukemia, others are representative of 3/group). **d** Manual differentials of peripheral blood from mice 5 weeks after tamoxifen induction ($n = 3$/group), $**p < 0.01$, $****p < 0.0001$. **e** Representative bone marrow histology (scale bar represents 10 μm). **f** Spleen weight of mice sacrificed at 5 weeks post tamoxifen treatment, $*p < 0.05$ as calculated by Student's $T$-test **g** Representative spleen histology (scale bar represents 100 μm for upper panels and 20 μm for lower panels). In all cases, values are represented as mean with error bars representing SEM. Source data are provided as a Source Data file.

order-dependent disease biology have not yet been discovered. If reprogramming of the lineage-specific enhancer repertoire is a common initiating event in AML, this creates even more impetus for the development of treatment strategies targeting these epigenetic pathways.

Our finding that epigenetic dysfunction is an obligate early event is likely to be generalizable to other forms of cancer. In renal cell carcinoma, loss of the tumor suppressor von Hippel-Lindau (VHL) is widely regarded as an initiating event and associated with dramatic changes in global DNA methylation, potentially impacting subsequently acquired signaling mutations[34]. In breast cancer, mutation order is an important determinant of cancer phenotype, with luminal-type tumors demonstrating early loss of PTEN while basal-type tumors display early p53 mutation[35]. In colon cancer, the canonical APC mutations decrease DNA methylation through upregulation of demethylases, which potentially alters the impact of RAS mutations acquired later in disease evolution[36]. Order-dependent mutational phenotypes were recently reported in myeloproliferative neoplasms, where the order of mutations in

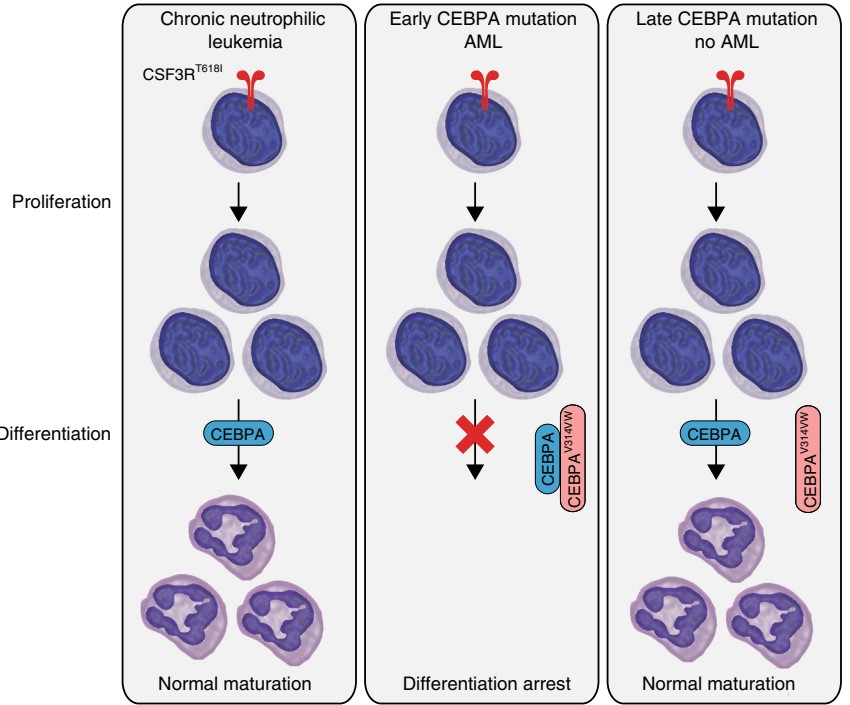

**Fig. 8** CEBPA mutations initiate AML through disruption of myeloid lineage enhancers. During normal myeloid differentiation, GCSF/CSF3R signaling drives both differentiation and proliferation of myeloid precursors. The differentiative program downstream of GCSF/CSF3R signaling is dependent on CEBPA acting at myeloid lineage enhancers. Mutant CEBPA blocks activation of these enhancers preventing transcription of differentiation-associated genes. When CSF3R mutations are introduced early, they activate transcription using the native enhancer repertoire initiating differentiation. Importantly this differentiation is insensitive to subsequent introduction of mutant CEBPA.

JAK2 and TET2 are important determinants of disease phenotype[37]. When TET2 precedes JAK2, patients are more likely to present with polycythemia vera. In contrast, patients with JAK2-first disease were more likely to have essential thrombocytosis. Thus, it appears that preceding TET2 mutation provides an epigenetic context supporting erythroid development for subsequently-acquired JAK2 mutations.

In summary, we describe the epigenetic mechanism by which mutant CEBPA and CSF3R interact to drive AML development. Our study demonstrates that a subset of differentiation-associated enhancers are dysregulated by mutant CEBPA, preventing normal myeloid maturation. Critically, this epigenetic dysregulation must occur as the initial event otherwise AML does not develop. These differentiation-associated enhancers represent a promising novel therapeutic target in this poor prognosis molecular subtype of AML.

## Methods
**Mice.** C57BL/6J mice (#000664), Balb/cJ mice (#000651), Rosa26 ERT$_2$-Cre mice (#008463), MX-1 Cre mice (#003556), and CEBPA$^{Flox/Flox}$ mice (#006447) were obtained from The Jackson Laboratories. Female mice were used between 6 and 20 weeks, and age and weight matched in all experiments. Tamoxifen was dissolved in corn oil (Sigma) and injected intraperitoneally at 75 mg/kg/day for 5 days. Poly I:C (Sigma) was dissolved in PBS at 12 mg/kg on days 1, 3, and 5. All experiments were conducted in accordance with the National Institutes of Health Guide for the Care and Use of Laboratory Animals, and approved by the Institutional Animal Care and Use Committee of Oregon Health & Science University (Protocol #TR01_IP00000482).

**Cell Lines.** 293T17 cells were obtained from ATCC and cultured in DMEM (Gibco) with 10% fetal calf serum (FCS; HyClone). Murine HoxB8-ER cells were a generous gift from David Sykes (Massachusetts General Hospital, Boston, MA) and cultured in RPMI (Gibco) with 10% FCS and CHO-SCF cell conditioned media (final concentration ~100 ng/mL)[27]. Wild-type HoxB8-ER cells were differentiated in estrogen-free media[23]. Cell lines were tested monthly for mycoplasma

contamination and all lines utilized tested negative. Cell lines were authenticated by ATCC prior to shipment.

**Cloning and Retrovirus production.** The following plasmids were utilized: pMSCV-IRES GFP[9], pMXs-IRES-Puro (Cell Biolabs Inc), pMSCV-IRES-mCherry FP (a gift from Dario Vignali, Addgene #52114), pMSCV-loxp-dsRed-loxp-eGFP-Puro-WPRE (a gift from Hans Clevers, Addgene #32702). The CSF3R$^{T618I}$ mutation was generated previously[9]. Full-length CEBPA and JAK3 cDNA was obtained from Genecopia. The CEBPA and JAK3 mutations were generated using the Quikchange Site Directed Mutagenesis Kit (Agilent) using the primers in Supplementary Table 1. To produce retrovirus, 293T17 cells were transfected with EcoPac helper plasmid (a gift from Dr. Rick Van Etten) and the appropriate transfer plasmid. Conditioned media was harvested 48–72 h after transduction.

**Western blotting.** K562 cells were transfected with lipofectamine 2000 according to the manufacturer's instructions. Normal mouse bone marrow cells were isolated fresh on the day of the experiment. CSF3R$^{T618I}$/CEBPA$^{V314VW}$ cells were cultured in IMDM media supplemented with 20% FCS after isolation from methocult. In all cases, $2 \times 10^6$ cells were used per condition. Cells were lysed in cell lysis buffer (Cell Signaling Technologies) containing complete mini protease inhibitor tablets (Roche). To pellet cellular debris, the lysates were spun at 12,000 r.p.m., 4 °C for 10 min, and subsequently mixed with 3× SDS sample buffer (75 mmol/L Tris (pH 6.8), 3% SDS, 15% glycerol, 8% β-mercaptoethanol, and 0.1% bromophenol blue). Samples were incubated at 95 °C for 5 min and run on Criterion 4 to 15% Tris-HCl gradient gels (Bio-Rad). Gels were transferred to PVDF membranes and blocked in Tris-buffered saline with Tween (TBST) with 5% bovine serum albumin (BSA). Blots were probed with the antibodies listed in Supplementary Table 3. Horseradish peroxidase-conjugated secondary antibodies against mouse IgG and rabbit IgG (CS) were used followed by imaging of the blots on a BioRad Touch Gel Doc (BioRad).

**Retroviral transduction.** Retroviral conditioned media was produced using 293T17 cells transduced with packaging plasmid and the appropriate transfer plasmid. Mouse bone marrow cells were harvested from healthy syngeneic donors at age 6–12 weeks and spinnoculated on two successive days with supernatant in the presence of polybrene[9,16]. Cells expressing GFP or GFP/mCherry were sorted by FACS using an FACSAria III sorter (BD). For colony assay, 2500 sorted cells were used per replicate. For bone marrow transplantation experiments, donor mice were treated with 100 mg/kg 5-FU 5 days prior to harvest. In all cases, 200,000 cells were administered, with the balance of non-transduced cells as fresh syngeneic

bone marrow. Survival endpoints for mutation comparison studies included WBC count >100, weight loss >20% initial weight, and moribund appearance.

**Real-time PCR**. RNA was extracted using RNeasy or RNeasy micro kit (Qiagen). cDNA was synthesized using a High Capacity cDNA Synthesis kit (ThermoFisher). Real-time PCR was performed using the QuantStudio7 Real Time PCR System (ThermoFisher) and Taqman primer probes (ThermoFisher). For Taqman low-density arrays, array cards were custom printed by the manufacturer (ThermoFisher). Taqman primer probes used in this study are listed in Supplementary Table 2.

**Flow cytometry**. The antibodies listed in Supplementary Table 3 were utilized for FACS according to the manufacturer instructions. Stained cells were analyzed on a FACSAria III flow cytometer (BD).

**RNA sequencing**. Bone marrow from C57BL/6 mice was harvested and retro-virally transduced with the indicated oncogene combinations as described above. Cells were stained with a lineage cocktail, and lineage-negative GFP/mCherry-double-positive cells were sorted. For order of acquisition experiments, cells were cultured in cytokine-free I20 media for 48 h after sorting in the presence of 500 nM 4-Hydroxytamoxifen. RNA was extracted using the RNeasy micro kit (Qiagen). cDNA libraries were constructed using the SMARTer universal low input RNA kit (Clonetech) and sequenced using a HiSeq 2500 Sequencer (Illumina) 100 bp SR.

**Microarray**. Labeled target, sscDNA was prepared from 12 RNA samples using the GeneChipTM WT-Plus protocol (ThermoFisher). Prior to cDNA synthesis and amplification, sample order was randomized. Amplified and labeled cDNA target samples were hybridized to a GeneChip Clariom S-Mouse array (ThermoFisher). Image processing was performed using Affymetrix Command Console (AGCC) v.3.1.1 software and expression analysis was performed using Affymetrix Expression Console. Differential expression analysis was performed using the Transcript Analysis Console 4.0 (ThermoFisher).

**Chromatin-immunoprecipitation sequencing (ChIP-seq)**. Twenty-five million cells per condition (Empty vector, CSF3R$^{T618I}$-only, CEBPA$^{V314VW}$-only, and CSF3R$^{T618I}$ + CEBPA$^{V314VW}$) were fixed with 1% formaldehyde and quenched by glycine. Cells were resuspended in lysis buffer (0.1% SDS, 0.5% Triton X-100, 20 mM Tris-HCl pH = 8.0, 150 mM NaCl, 1× Proteinase inhibitor (Roche)), and sheared using the Bioruptor Pico sonicator (Diagenode). Reactions were rotated overnight at 4 °C with antibody (H3K4me1 (ab8895; Abcam), H3K4me3 (ab8580; Abcam), and H3K27ac (ab4729; Abcam), Table S10). Next, samples were rotated with Protein A/G Magnetic Beads (Pierce). Beads were washed with TBST buffer (3×), lysis buffer, and 2× TE pH = 8. Chromatin was eluted at room temperature in 1% SDS, 0.1 M NaHCO₃. Crosslinks were reversed at 65 °C overnight, digested with RNAse A and proteinase K, then purified by phenol–chloroform extraction. Sequencing libraries were generated using the NEBNext Ultra II DNA Library Prep Kit for Illumina (New England Biolabs). Libraries were sequenced using SE 75 bp Illumina NextSeq.

**RNA-seq analysis: murine samples**. Raw reads were trimmed with Trimmo-matic[38] and aligned with STAR[39]. Differential expression analysis was performed using DESeq2 (ref. [40]). Raw $p$ values were adjusted for multiple comparisons using the Benjamini–Hochberg method.

**RNA seq analysis: TARGET pediatric AML samples**. RNA sequencing was performed on RNA collected from pediatric AML samples as described in the initial TARGET AML publication[18]. Raw reads were aligned with Kallisto with count tables produced using Tximport[41,42]. Differential expression analysis was performed using DESeq2 (ref. [40]). Raw $p$ values were adjusted for multiple comparisons using the Benjamini–Hochberg method. For comparison with mouse RNA seq, genes with differential expression between CSF3R WT and CSF3R$^{T618I}$ were mapped to orthologous human genes using ensembl BioMart. Genes with differential expression in both mouse and human were considered and compared by Log2 Fold change between CSF3R$^{mutant}$ and CSF3R$^{WT}$ conditions.

**Convergent gene expression analysis: Leucegene AML**. RPKM values from adult AML samples from the Leucegene cohort were downloaded using the following SRA accession numbers: GSE49642, GSE52656, GSE62190, GSE66917, GSE67039. Mouse genes with differential expression driven by CSF3R$^{T618I}$ were converted to human gene symbols as above. Genes demonstrating an absolute fold change of >2 in both datasets were compared by Log2 Fold change between CSF3R$^{mutant}$ and CSF3R$^{WT}$ conditions.

**Motif enrichment analysis**. The enrichment analysis for motifs was performed using HOMER[43] using the -findMotifs command in a 500 bp window upstream and 200 bp downstream of the transcriptional start site. De novo motifs at enhancers were identified using the -findMotifsGenome command in HOMER in a

±1000 bp region surrounding the peak center. De novo motifs were matched to their closest known motif and displayed with the alignment score (with 1 being a perfect match). Top five motifs with $p < 1E-10$ for each group of promoters or enhancers are displayed.

**Gene set enrichment analysis**. Gene set enrichment analysis using the GSEA software[44]. As CSF3R$^{T618I}$-only and CSF3R$^{T618I}$-first groups clustered together by Euclidian distance, they were combined and compared with CEBPA$^{V314VW}$-first samples. Analysis was performed using the C2 collection from MSigDB. Permutations were performed by gene set and significance was set as an FDR adjusted $p$ value of <0.05.

**Permutation analysis**. CEBPA ChIP-seq peaks from GMPs[17] were converted from the mm9 to mm10, using the liftOver tool from the UCSC genome browser. We used BEDTools[45] to randomly shuffle the location of all interacting genes, within their original chromosomes. After each shuffle, we used the closest-features sub-command in BEDOPS[46] to calculate the distance between the new position of all shuffled genes and the closest CEBPA ChIP-seq.

**ChIP-seq analysis**. Reads were aligned to the mouse reference genome (mm10) using bwa 0.7.12 (ref. [47]) with default single end settings. Low mapping alignments were removed with samtools (MAPQ <30)[48]. For quality control purposes, PCA was performed using Deeptools[49] which showed separation of signal by oncogene condition. Next, MACS2 2.1.1 (ref. [50]) was used to predict significant peaks of ChIP-seq enrichment relative to the appropriate input controls and generate fold enrichment tracks. We used the ChromHMM software[51] to characterize and annotate the genomes of each treatment group according to six chromatin states, based on different combinations of H3K4me1, H3K4me3, and H3K27ac marks. Enhancers were identified through the presence of H3K4me1 and H3K27Ac, absence of H3K4me3. Enhancers less than 500 bp apart were merged. The closest gene to each enhancer was identified using BEDOPS[46]. In order to remove potential false positives due to proximity to large H3K27ac peaks, we removed treatment-specific putative enhancers without a H3K27ac peak summit.

**Gene Ontology Analysis for ChIP-seq**. Gene Ontology Analysis for histone mark ChIP-seq performed by Genomic Regions Enrichment of Annotations Tool[52] using the basal plus extension model to annotate enhancer coordinates with nearby genes.

**Correlation of gene expression with enhancer activation**. All differentially expressed genes as assessed by microarray analysis utilized for unsupervised hierarchical clustering analysis with a k-means = 8. Each active enhancer was then annotated to the nearest gene. Condition-specific enhancers associated with each cluster of differentially expressed genes were counted and enrichment assessed as described under statistics.

**Overlap of CEBPA peaks with condition-specific enhancers**. We used publicly available genome-wide positions of all GMP CEBPA ChIP-seq peaks[17]. CEBPA peaks overlapping each group of condition-specific enhancers were identified using the mergePeaks command in HOMER with the -cobound option. CEBPA peaks associated with each group of condition-specific enhancers were counted and enrichment assessed as described below under statistics.

**Enhancer overlap using BEDTools**. Published human CEBPA AML DNase sensitivity peaks[27] and enhancers[22] were compared with mouse CSF3R/CEBPA-specific enhancers. The coordinates for mouse enhancers converted to human coordinates using the UCSC liftover tool. Overlap was assessed using the BEDTools fisher test[45]. Enrichment of overlap is detected using a Fisher's exact test with a significant right-sided $p$ value while depletion is detected with a significant left-sided $p$ value.

**Quantification and statistical analysis**. Data are expressed as mean ± SEM. Statistical analysis was performed using Prism software (Version 7.0; Prism Software Corp.) or RStudio. Statistical analyses are described in the figure legends. All data were analyzed with either an unpaired Student's $t$-test, or ANOVA followed by post hoc analysis using a Sidak's corrected $t$-test. For Taqman-based array, microarray and RNA-seq data, $p$ values were adjusted for repeated testing using a false discovery rate by the method of Benjamini–Hochberg[53]. For enhancer enrichment analyses, a $\chi^2$ analysis was utilized with individual $p$ values adjusted by the method of Holm–Bonferroni. Survival analysis was conducted using the method of Kaplan–Meier and statistical significance was assessed using a log rank test.

**Reporting summary**. Further information on research design is available in the Nature Research Reporting Summary linked to this article.

## Data availability

The accession numbers for all genomic data reported in this paper is GSE122166. The source data underlying Figs. 1–7 and Supplementary Figs. 1–6 are provided as a Source Data file.

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

## Acknowledgements

We are grateful to the following core facilities: Histopathology Shared Resource, Massively Parallel Sequencing Shared Resource, Gene Profiling Shared Resource, Epigenetics Consortium, Knight Cancer Institute Biostatistics Shared Resource, Flow Cytometry Shared Resource, ExaCloud Cluster Computational Resource and the Advanced Computing Center. Funding provided by an American Society of Hematology Research

Training Award for Fellows and Collins Medical Trust Award to T.P.B., OHSU School of Medicine Faculty Innovation Fund to B.J.D. and J.E.M., Howard Hughes Medical Institute to B.J.D, NCI R00-CA190605, an ASH Scholar Award, and an MRF New Investigator Grant to J.E.M. NCI U10 CA 98543-08S6, a St. Baldrick's Foundation Grant and Help the Hutch Funding to S.M., L.C. and M.O are supported by the National Science Foundation (grant #1613856) and from the National Institute of Health (R01HG010333), L.C. is also supported in part by NIH/OD P51 OD011092 to the Oregon National Primate Research Center.

## Author contributions

Conceptualization: T.P.B., B.J.D., and J.E.M. Methodology: T.P.B., M.O., and J.E.M. Software: T.P.B., M.O., B.D., S.J., and S.M. Formal analysis: T.P.B, M.O., B.D., B.R.W., M.R.G, J.C.E., H.G.M., K.W., J.L.S., A.R.L., R.E.R., and S.J. Investigation: T.P.B., C.C., S.A.C., A.F., Z.S., K.N., B.M.C., D.L., B.G., C.D., R.D., and J.E.M. Data curation: T.P.B., M.O., B.D., S.J., and S.M. Writing—original draft: T.P.B., L.C., and J.E.M. Writing—review & editing: T.P.B., M.O., L.C., B.J.D., and J.E.M. Supervision: T.P.B., S.M., L.C., B.J.D. and J.E.M. Funding acquisition: T.P.B., C.N., S.M., L.C., B.J.D., and J.E.M.

## Competing interests

B.J.D. potential competing interests—Consultant: Monojul, Patient True Talk; SAB: Aileron Therapeutics, ALLCRON, Cepheid, Gilead Sciences, Vivid Biosciences, Celgene & Baxalta (inactive); SAB & Stock: Aptose Biosciences, Blueprint Medicines, Beta Cat, GRAIL, Third Coast Therapeutics, CTI BioPharma (inactive); Scientific Founder & Stock: MolecularMD; Board of Directors & Stock: Amgen; Board of Directors: Burroughs Wellcome Fund, CureOne; Joint Steering Committee: Beat AML LLS; Clinical Trial Funding: Novartis, Bristol-Myers Squibb, Pfizer; Royalties: OHSU #606-Novartis exclusive license, OHSU #2573; Dana-Farber Cancer Institute #2063-Merck exclusive license. The remaining authors declare no competing interests.
