## [Peer Review File · Nature Communications]

Reviewers' comments:

Reviewer #1 (Remarks to the Author):

In this study the investigators developed mouse models to determine the transforming abilities of mutated CSF3R (activating mutations) in combination with mutated CEBPA (loss of function mutations) when introduced in mouse bone marrow cells via retroviral transduction. These two mutations have been observed recurrently in primary AMLs. In vitro and in vivo experiments revealed full blown leukemia and block of myeloid differentiation when the two mutated genes were combined in mouse bone marrow cells. The authors also provide data that suggest that CEBPA mutations have to occur first, as marrow cells with first the CSF3R mutant do not develop AML upon mutant CEBPA introduction. My comments and concerns are the following:

In general, I find the paper not always easy to read and the order of paragraphs not easy to follow. In particular the connection between the different paragraphs is not always clear to me. Sometimes it feels as if different small sets of experiments have been performed with the bone marrow cells of the mice and these different "little projects" have been put together as one paper

My following two comments are about the concept and the approach the investigators followed in this study. CEBPA loss of function- and CSF3R activating mutations indeed co-occur infrequently but recurrently in primary AML in humans. Generating a mouse model to understand mechanisms of collaboration is useful. There are two important differences between AML and the mouse models used to study co-operation, which are important to judge the study on its impact.

- In humans the loss-of-function mutations almost always occur together with an N-terminal mutation at the other allele. So no wild-type p42 CEBPA is expressed in those leukemias. In the mouse model, the investigators introduced mutant CEBPA retrovirally into the bone marrow and studied the effects of it in vitro and in vivo. In fact they studied the effects of mutant CEBPA in the context of the presence of wt p42 CEBPA. The levels of mutant versus wt-CEBPA are also not determined. Conceptually loss-of-function CEBPA together with wt-CEBPA in one cell is different from a cell with loss-of-function CEBPA and N-terminal mutant CEBPA and thus no wt-protein. Maybe the investigators have certain explanation why they carried out the experiments in the way presented here, but one would at least expect a proper discussion about the differences between the model and the patient situations and explain why they carried out the study, the way they did. Moreover, the outcome of the study should also be discussed within this context.
- The second issue I wish to discuss is equally important. One of the most important targets of CEBPA is the gene encoding CSF3R. Loss of CEBPA, as has been shown by multiple investigators, causes down-regulation or even loss of expression of CSF3R. Thus the two genes that are mutated in AML are not representing totally different pathways, but CEBPA wt or mutant, affect the expression of mutated CSF3R! This point is totally ignored in the results and in the discussion. In fact, the authors state in the Results section, "We confirmed that neither oncogene was changing the expression of the other...". This may indeed be true, but this is not what happens in the natural situation and in AML with these mutations.

A model is a model and may not fully represent a particular situation, but I find it a major concern that this has not at all been taken into account either in the set-up of the study or at least in the discussion of the work.

In general the colony forming data presented in Figure 1 are nice and show nicely that cells form colonies independent of cytokines when the mutant CSF3R is expressed. The effects of introducing mutant CEBPA to CSF3R-mutant cells is also convincing. What I miss is nice morphology of the colony cells. It should show mature cells if only mutant CSF3R is expressed compared to mainly blasts in double mutant or only CEBPA mutant cells.

I find the part of gene expression profiling not easy to read and I really wonder what we are looking at here. We look at signatures of total bone marrow samples of marrow samples with only blasts (when mutant CEBPA is expressed) versus marrow with many differentiated myeloid cells

(When only mutant CSF3R is expressed). The differences are now a mixture of altered gene expression directly related to CEBPA or CSF3R mutated and differences in maturation of the marrows. Most likely the differentiation is the major component driving differences, as the authors themselves indicate that they found multiple "neutrophil" genes differentially expressed. These type of gene expression experiments will have a meaning when purified progenitors are isolated and studied by RNA-Seq. Single cell RNA-Seq would even be better.

The authors conclude that due to loss of function of CEBPA mutations altered enhancer activation is found. These conclusions are all based on presence or absence of histone marks and yes or no presence of CEBPA binding sites in certain enhancers. CEBPA ChIP-Seq data of WT versus mutant are essential for such conclusions. This is not done or even considered. In the discussion this omission is not mentioned. The question is whether this is possible using the model. How to discriminate between wt and mutant CEBPA? ChIP-Seq data of primary AMLs for CEBPA would have been appropriate.

Reviewer #2 (Remarks to the Author):

Braun et al. study the cooperation between CSF3R and CEBPA mutations in AML using retroviral transduction of mouse bone marrow cells and cell lines. Transcriptional (RNA-seq, microarray) and epigenetic (ChIP-seq) analysis informs genes and regulatory elements that are affected. It is convincingly demonstrated that CSF3R activation and CEBPA mutations cooperate to cause leukemia. Activating the JAK/STAT pathway and inhibiting CEPBA activity through other means has similar effects, which implies that various genetic lesions converge on a limited number of pathways in AML. It is interesting and to my knowledge novel that the order of these mutations determines the phenotype of the leukemia. For the most part, the work employs appropriate models and analyses, but I suggest numerous opportunities for improvements and additional controls below. Overall, the paper is a valuable contribution to the field of AML, with concepts that may apply to other tumors.

COMMENTS

1. The relationship between AML-ETO, RUNX1, core binding factor AML and CEBPA is not common knowledge. It should be explained clearly, preferably in the introduction.
2. Several comments about the paragraph "To further describe the transcriptional changes...".
 - 2.1 "Of particular interest was a cluster of genes that were strongly up-regulated by CSF3R-T618I, but suppressed by co-expression of CEBPA-V314VW, as this mirrored the pattern of myeloid differentiation." The authors imply that CSF3R-T618I alone increases myeloid differentiation. Is that supported by functional assays in Figure 1?
 - 2.2 Figure 2D: Why are there eight heatmaps? Are they distinguished by K-means?
 - 2.3 What is the rationale for calling the first cluster "interacting genes"?
 - 2.4 How do the five columns in Figure 2E (and S1B) correspond to the eight heatmaps in Figure 2D? It may be clearer to use the same five categories in 2D.
 - 2.5 "60% of the interacting genes were within 2 Kb of a CEBPA peak, compared with approximately 20% of non-interacting genes." Please do this analysis for each of the five (or eight) categories. Considering you focus on enhancers later, 2 kb is a very small window.
3. "Interestingly, CSF3R mutant samples tended to display increased HOX gene expression although this did not rise to the level of statistical significance." The legend of Figure 1F appears to be wrong. Assuming red is double mutant, it looks like there is no difference with the purple dots. This point can (and should) just be omitted.
4. "Comparison of CEBPA mutant/CSF3R WT and CEBPA mutant/CSF3R mutant patient samples

revealed numerous differentially-expressed genes (Table S2). Principle component analysis using these differentially-expressed genes clearly separates CEBPA/CSF3R mutant samples from those harboring mutant CEBPA alone (Figure S1D)."

It is "Principal". More consistency in naming groups would improve comprehensibility. Most importantly, it is unsurprising that PCA using differentially expressed genes between two groups is going to separate those groups, this is circular.

5. In Figure 3, the authors show in vivo cooperativity between JAK/STAT and CEBPA pathways. This is very interesting. It would be more convincing with additional controls. In 3A, why have the authors not done CEBPA-V314VW alone? And why is the survival of CSF3R-T618I alone worse than the survival of CSF3R-T618I + CEBPA-F82fs? In Figure 3D, what does a healthy control look like? In Figure 3K, what does AML-ETO alone look like?

6. Figure S5D-S5E. The authors argue overlap between enhancers indicates conserved biology. How about a negative control, for example, there should be no enrichment when using human MLL AML enhancers or peaks.

7. "We observed strong enrichment of CSF3R-T618I-specific enhancers in Clusters 1 and 4, which displayed the highest level of gene expression with CSF3R-T618I alone." Please add interpretation. It seems to me that Figure 4F shows that open enhancers are associated with increased gene expression.

8. The paragraph about enhancer priming and H3K4me1 intensity is difficult to understand. The first statement ("H3K4me1 enrichment ... CEBPA-V314VW only") is circular and the second statement ("Regions with specific enhancers") is not supported by any figure or statistical analysis. It would be better to omit this paragraph. The poorly supported conclusion that the mutations affect H3K4me1 is not vital to the paper. Figure 4H-I on the other hand are great.

9. Figure 5E: Have the authors done CEBPA only?

10. Figure 5G, Figure S6: Why did the authors pool results for CSF3R First + CSF3R Only? It seems a better comparison would be CEBPA First vs. CSF3R First.

11. The last section about the order of mutations is interesting. Is it possible to compare AML patients with CSF3R VAF > CEBPA VAF to AML patients with CSF3R VAF < CEBPA VAF (classification, outcome, ...)?

12. Figure 7: It is great to have a graphical summary for clarification. However I find this figure confusing. Can you find a way to integrate some aspects of the attached figure?

MINOR COMMENTS

13. In the abstract, it should be "applicable TO other".

14. Introduction: In the first sentence, "genetic" would be a better term than "genomic". CEBPA is "a", not "the", master regulator of myeloid lineage commitment. What is the frequency of CSF3R and CEBPA mutations in AML and the overlap?

15. Results: "sustained growth in liquid media lacking cytokines (Figure 1C)." That is not what Fig. 1C shows.

16. The columns in Figure 2B-C-D should be in the same order.

17. As everybody knows, you have to describe figures in order. In the text, S1C is described before S1B. Similarly, Figure 2I is described before 2H. Similarly, Figure S5D-E are described

before Figure S5A-C. Similar issues with Figure 6.

18. Discussion: "which no doubt alter the impact of RAS mutations acquired later in disease evolution" is a very strong statement.

19. Figure S4C: Please add Empty Empty.

20. Figure S5D: the legend is incomplete.

21. Figure S7E: why does it say Week?

22. It is not clear which supplementary table is which. Please add the table name in first row.

Reviewer #3 (Remarks to the Author):

The authors use retroviral over-expression in murine bone marrow cells to study interaction between CEBPA and CSF3R mutations in AML. They conclude that silencing of critical myeloid enhancers is important for the functional co-operation between these genetic lesions. Overall, this is an interesting study with an impressive amount of data. I have a number of specific comments. Major comments:

1. The level of over-expression of the various CSF3R and CEBPA constructs in mouse bone marrow cells should be assessed by western blot (relevant to Figs 1 and 5). As the authors are using retroviral over-expression, these experiments are essential to interpret all the subsequent data. N-terminal CEBPA mutations typically result in expression of a truncated 30kDa isoform from an internal initiation site (PMID11242107) – can this be confirmed for the CEBPAF82fs construct? Likewise, the impact of mutant CSF3R on downstream signaling should be analyzed.

2. In many published patient cohorts, bi-allelic CEBPA mutations are more frequent than mono-allelic mutations and have distinct transcriptional, epigenetic and prognostic features (e.g. PMIDs 27276561 and 28408400). Collectively there is a large body of work that argues that CEBPA mutations are functionally distinct from loss of CEBPA expression. It is not entirely clear what the authors are attempting to model with their over-expression strategy (presumably mono-allelic mutations). To their credit, they do attempt to reconcile gene expression data from their model with data from AML patients. However, I would suggest: (1) the issue of mono-allelic vs bi-allelic CEBPA mutations should be dealt with in the introduction; (2) the caveats of the modelling approach should be outlined in the discussion; and (3) when analysing patient data, it should be clarified whether all CEBPA mutant patients are being considered together (or whether mono-allelic and bi-allelic patients are considered as separate entities).

3. The authors propose a model whereby mutant CEBPA displaces wild-type CEBPA from enhancers formed downstream of CSF3R mutation (Fig.7), however, this is largely based on motif enrichment. Direct evidence for this proposed model by way of CEBPA ChIPseq (or at least ChIP-PCR at a panel of target enhancers) would strengthen the study (and should be achievable in HoxB8 cells).

4. Related to the above point, I am unclear how the model explains the functional co-operation between CSF3R and CEBPA mutations. It partially explains the block of differentiation (failure to activate enhancers important for differentiation) but if wild-type CEBPA is required for mutant CSF3R activity then CEBPA loss of function and CSF3R gain of function mutations would be antagonistic. Are there genes/enhancers that are uniquely activated in the CSF3RT618I/CEBPAV314VW double mutant cells that may explain the phenotype? Or is it perhaps that the differentiation-associated CSF3RT618I-gene expression programs are CEBPA-dependent but the proliferation-associated programs are not? I think some more detail is required here. Functional validation of key target genes may also be appropriate.

Minor comments:

1. The authors compare active enhancers identified specifically in CSF3RT618I/CEBPAV314VW

HoxB8 cells with enhancers in CEBPA mutant patients and find extensive overlap (Figs S5D and S5E). This seems to go against their argument that CSF3R and CEBPA mutations co-operate functionally to alter the enhancer landscape in a manner that is more than additive. Can the authors please provide an explanation?

2. The authors use an elegant system to test the importance of mutation order for the phenotype and conclude that "when CEBPA mutations are introduced after mutations in CSF3R, they are unable to fully block myeloid differentiation". To fully justify their conclusion the authors would need to show (e.g. by FACS) that at the time when they induce mutant CEBPA expression in CSF3RT618I-first cells, the cells are not already differentiated (relative to an empty vector control). I think that a more likely explanation is that that loss of CEBPA activity cannot reverse differentiation.

3. Related to the above point, analysis of enhancers in the sequential model (e.g. ChIP for H3k27ac in the CEBPA-first cells versus the CSF3R-first cells) could prove really informative

4. Figure 4C does not seem to be referenced in the text.

5. There appears to be something wrong with the PCA plot in Figure S5C with no variation between samples in the first principle component

6. Scales/labels are missing on some figures:

- Fig 5E: scale label

- Fig S1A: missing key for point size (p value?)

- Fig S5A: scale labels, y axis metagene plot label missing

7. "morphologic neutrophils in the peripheral blood and bone marrow (Figures 2A-B..." I think is referring to Figures S2A-B)

Response to Reviewer 1:

In general, I find the paper not always easy to read and the order of paragraphs not easy to follow. In particular the connection between the different paragraphs is not always clear to me. Sometimes it feels as if different small sets of experiments have been performed with the bone marrow cells of the mice and these different “ little projects” have been put together as one paper

We appreciate the feedback from the reviewer regarding the flow of the manuscript. We agree that the connection between each piece of data, particularly within the first several figures was not always clear. We presented data from multiple mutational pairings in an effort to make our findings generalizable to more mutational subsets of AML than simply CSF3R/CEBPA mutant AML. However, we agree that this made the flow of the manuscript difficult to follow. To simplify and improve readability, we have removed data relating to AML-ETO as this particular aspect of the story was (as other reviewers point out) incompletely developed. This data was not crucial to the principle findings of the manuscript and removing it creates space to add additional description and experiments as suggested by the reviewers. Additionally, we have worked on the flow of the text to improve the transitions between paragraphs.

In humans the loss-of function mutations almost always occur together with an N-terminal mutation at the other allele. So no wild-type p42 CEBPA is expressed in those leukemias. In the mouse model, the investigators introduced mutant CEBPA retrovirally into the bone marrow and studied the effects of it in vitro and in vivo. In fact they studied the effects of mutant CEBPA in the context of the presence of wt p42 CEBPA. The levels of mutant versus wt-CEBPA are also not determined. Conceptually loss-of-function CEBPA together with wt-CEBPA in one cell is different from a cell with loss-of-function CEBPA and N-terminal mutant CEBPA and thus no wt-protein. Maybe the investigators have certain explanation why they carried out the experiments in the way presented here, but one would at least expect a proper discussion about the differences between the model and the patient situations and explain why they carried out the studied, the way they did. Moreover, the outcome of the study should also be discussed within this context.

We agree with the reviewer that mutations in CEBPA often occur as a compound heterozygous N- and C-terminal mutation and that this warrants additional discussion in the manuscript. However, this is not invariably the case and monoallelic CEBPA mutations are also found. In pediatric AML, where we first reported the co-occurrence of CSF3R and CEBPA mutations, 5 patients had mutant CSF3R and bi-allelic CEBPA mutations, while 3 patients had mutant CSF3R and only a C-terminal CEBPA mutation and one patient had mutant CSF3R and an N-terminal CEBPA mutation (Maxson et al, Blood, 2016). In the companion paper reporting on CSF3R/CEBPA mutant adult AML, one of the 4 CSF3R mutation patients had an atypical CEBPA mutation pattern (Lavalley et al, Blood 2016).

Our study was therefore conducted with this pattern of CSF3R and CEBPA mutations in mind. That being said, we completely agree with the reviewer that retroviral expression

of a mutant CEBPA in a cell expressing wild type CEBPA is quite different than what is seen in a large fraction of patients with co-occurring CSF3R and CEBPA mutations. To address this issue, we have made the following changes to the manuscript including data from an additional murine model:

- Through a collaboration with Claus Nerlov, we have obtained fetal liver hematopoietic cells harboring a compound heterozygous N and C terminal CEBPA mutation (Bereshchenko et al, Cancer Cell, 2009). When transplanted into syngeneic recipients these cells produce a myeloid leukemia over the course of approximately one year (cells have to be transplanted because the mouse has an embryonic lethal phenotype). Prior to transplantation, we transduced these fetal liver cells with mutant CSF3R. This experiment had been initiated prior to our initial submission, however owing to the long latency of disease development we did not include it with our first submission. However, it has sufficiently matured at this point and we are now pleased to include these data. They confirm our finding that mutant CSF3R accelerates disease development in the context of endogenous compound heterozygous CEBPA mutations (Please see Figure 3 A, B)
- As suggested by the reviewer we have added a discussion of N- and C-terminal CEBPA mutations and their patterns of co-occurrence to the manuscript on lines 69-105 in the introduction and lines 433-456 in the discussion.

“The transcription factor CCAAT enhancer binding protein alpha (CEBPA) is a master regulator of myeloid lineage commitment. CEBPA is recurrently mutated in AML and is a classic example of a Class II mutation². The CEBPA gene is comprised of a single exon with an internal translational start site. Mutations in CEBPA cluster at the N- and C- terminus of the protein. N-terminal mutations typically result in a frame shift, leading to a premature stop codon and loss of expression of the long (p42) isoform of the protein but ongoing translation of the short isoform (p30). The p30 isoform lacks a crucial transactivation domain that represses cell cycle progression through a direct interaction with E2F³. In contrast, C-terminal mutations occur in the DNA binding domain leading to loss of function and a blockade in granulocytic differentiation². The most common pattern of mutation in AML is an N-terminal mutation on one allele and a C-terminal mutation on the other allele. This results in alterations in the ratio of the p42 to p30 CEBPA isoforms, changing the balance of differentiation and proliferation. AML with biallelic CEBPA mutations is associated with favorable prognosis, with approximately 50% of younger patients achieving a cure with chemotherapy alone. However, the precise determinants of relapse in this mutational context are unknown.

Recent studies have identified a high rate of co-occurrence of mutations in CEBPA with mutations the granulocyte-colony stimulating factor receptor (CSF3R)⁴⁻⁷. Approximately 20-30% of patients with CEBPA mutant AML harbor a cooperating mutation in CSF3R. Although CSF3R mutations are often associated with biallelic CEBPA mutations, monoallelic cases also occur. Interestingly, monoallelic C-terminal CEBPA mutations are far more likely to co-occur with mutations in CSF3R. Patients with CSF3R/CEBPA mutant AML have inferior outcomes to those with mutant CEBPA alone, arguing that the presence of Class I mutations in CSF3R may be an important determinant of chemotherapy resistance and relapse⁸. The mutations in CSF3R most commonly occur

in the membrane proximal region, and lead to ligand independent receptor dimerization and constitutive signaling via the JAK/STAT pathway. Similar to other Class I mutations, membrane proximal CSF3R mutations produce a myeloproliferative phenotype when present in isolation and are the major oncogenic driver of the disease chronic neutrophilic leukemia⁹. During normal myeloid development, G-CSF signaling via CSF3R leads to proliferation of myeloid precursors and neutrophilic differentiation. CEBPA is required for CSF3R-mediated transcription of myeloid specific genes, and myeloid differentiation arrests at the level of the common myeloid progenitor when CEBPA is deleted^{10,11}. In spite of the established functional interdependence of CSF3R and CEBPA during normal hematopoiesis, the mechanism by which oncogenic mutations in these two genes interact to drive AML remains unknown.”

“N- and C-terminal CEBPA mutations exert distinct biological roles in leukemia initiation^{21,28}. While it is clear that reduced CEBPA activity is potentially oncogenic, CEBPA knockout mice fail to develop AML²⁹. Similarly, mice with homozygous C-terminal mutations (putatively loss of function) develop leukemia with an exceedingly long latency²¹. Interestingly, the phenotype of these leukemias is erythroid rather than myeloid. This is consistent with data from MLL-rearranged leukemias, where CEBPA is requisite for entrance into the myeloid lineage and leukemia initiation^{30,31}. As the majority of patients with CEBPA mutant AML harbor combined N- and C-terminal CEBPA mutations, it is possible that the N-terminal CEBPA mutation provides sufficient residual myeloid differentiation potential to enter the myeloid lineage and initiate AML. This relationship becomes even more complex when CSF3R mutations are considered. While the majority of CSF3R mutations occur in CEBPA-bi cases there are a number of CSF3R mutant CEBPA C-terminal monoallelic cases as well⁵. Our data demonstrate that CSF3R^{T618I} potentially synergizes with biallelic CEBPA mutations to induce AML. Our retroviral model best represents monoallelic cases (with the caveats of ectopic expression), and demonstrates that C- but not N-terminal CEBPA mutations exhibit synergy with mutant CEBPA confirming the high degree of clinical association between these mutations. It is possible that mutant CSF3R provides both a myeloid commitment signal and proliferative advantage, thus rendering N-terminal CEBPA mutations less crucial for AML initiation. Going forward, it will be crucial to define the biological interaction with mutant CSF3R and N- and C-terminal mutations present in the endogenous to understand whether these mutant forms of CEBPA demonstrate differential binding or recruitment of cofactors to critical differentiation-associated enhancers.”

- We have added an evaluation of the expression levels of the N- and C-terminal CEBPA mutants utilized by western blot (Supplemental Figure 1A).

The second issue I wish to discuss is equally important. One of the most important targets of CEBPA is the gene encoding CSF3R. Loss of CEBPA, as has been shown by multiple investigators, causes down-regulation or even loss of expression of CSF3R. Thus the two genes that are mutated in AML are not representing totally different pathways, but CEBPA wt or mutant, affect the expression of mutated CSF3R! This point is totally ignored in the results and in the discussion. In fact, the authors state in the

Results section, “We confirmed that neither oncogene was changing the expression of the other...”. This may indeed be true, but this is not what happens in the natural situation and in AML with these mutations. A model is a model and may not fully represent a particular situation, but I find it a major concern that this has not at all been taken into account either in the set-up of the study or at least in the discussion of the work.

We agree with the reviewer that this is an important point given the established cross regulation of CSF3R and CEBPA. To evaluate this, we investigated the expression of CSF3R in the TARGET pediatric AML cohort. The CSF3R gene was robustly expressed in the majority of cases and showed no significant pattern of regulation when mutant CEBPA was present. Thus, although as the reviewer points out, CEBPA mutations have been shown to dramatically downregulate CSF3R expression in other systems, this does not appear to be the case in human AML. Furthermore, if mutant CEBPA eliminated the expression of mutant CSF3R, it would be expected that this oncogene pairing would not alter disease biology. Interestingly, it has been recently reported that the presence of mutant CSF3R reverses the favorable prognosis associated with CEBPA mutant AML (PMID:31041512). Thus, it seems probable that mutant CSF3R must be expressed in the context of mutant CEBPA in order to impact disease biology and prognosis.

- We have added a figure demonstrating the expression of CSF3R is not significantly changed in CEBPA mutant versus CEBPA wildtype AML (Supplemental Figure 1I).
- We have added a discussion of this important point to the manuscript lines 227-233).

“As CEBPA is known to regulate CSF3R expression, we examined CSF3R expression in CEBPA WT and Mutant AML samples (Figure S1I)²⁰. Both subsets of patients robustly expressed CSF3R and no significant difference in expression was observed on the basis of CEBPA mutational status. Thus, although wild type CEBPA does regulate the activity of the CSF3R promoter, it appears that CSF3R expression is not dramatically altered in AML by the presence of putative loss-of-function mutations in CEBPA.”

In general the colony forming data presented in Figure 1 are nice and show nicely that cells form colonies independent of cytokines when the mutant CSF3R is expressed. The effects of introducing mutant CEBPA to CSF3R-mutant cells is also convincing. What I miss is nice morphology of the colony cells. It should show mature cells if only mutant CSF3R is expressed compared to mainly blasts in double mutant or only CEBPA mutant cells.

- This is an excellent suggestion, we have added cytopsin images from bone marrow cells transduced with varying combinations of mutations to Figure 1E. Indeed, we find that mutant CSF3R leads to expansion of more mature myeloid progenitors while mutant CEBPA produces blasts.

I find the part of gene expression profiling not easy to read and I really wonder what we are looking at here. We look at signatures of total bone marrow samples of marrow samples with only blasts (when mutant CEBPA is expressed) versus marrow with many differentiated myeloid cells (When only mutant CSF3R is expressed). The differences are now a mixture of altered gene expression directly related to CEBPA or CSF3R mutated and differences in maturation of the marrows. Most likely the differentiation is the major component driving differences, as the authors themselves indicate that they found multiple “neutrophil” genes differentially expressed. These type of gene expression experiments will have a meaning when purified progenitors are isolated and studied by RNA-Seq. Single cell RNA-Seq would even be better.

We apologize for any confusion caused by the way in which we had presented the data. For the RNA sequencing studies in Figure 2 and Figure 6, we transduced whole bone marrow but then FACS sorted lineage negative bone marrow transduced with both oncogenes (GFP and RFP). We did consider the approach suggested by the reviewer, however it is well established that progenitors differentiate in ex vivo culture and we were concerned that our mutations would influence this differentiation during the 3 days of culture necessary for efficient retroviral transduction. We also considered transducing whole bone marrow and sorting ckit positive progenitors receiving both oncogenes. However, we found that ckit expression is rapidly lost during the culture period. Thus, we felt that sorting lineage negative cells was the best way to control perturbations in the mixture of cells as the reviewer points out. Interestingly, rather than observe a decrease in the percentage of lineage negative cells, we found that **mutant CSF3R actually drove an increase in the size of the lineage negative population**, consistent with promoting stem cell proliferation.

- We have added figure panels clearly depicting our progenitor flow sorting strategy to Supplemental Figure 1B, C and Supplemental Figure 6A, B to make the experimental setup clear.

The authors conclude that due to loss of function of CEBPA mutations altered enhancer activation is found. These conclusions are all based on presence or absence of histone marks and yes or no presence of CEBPA binding sites in certain enhancers. CEBPA ChIP-Seq data of WT versus mutant are essential for such conclusions. This is not done or even considered. In the discussion this omission is not mentioned. The question is whether this is possible using the model. How to discriminate between wt and mutant CEBPA? ChIP-Seq data of primary AMLs for CEBPA would have been appropriate.

We appreciate the reviewer's suggestion and agree that CEBPA ChIP seq would be a valuable addition to the manuscript. Unfortunately, the CEBPA antibody used for ChIP seq by the majority of labs (Santa Cruz 14AA) has been discontinued. At the outset of this project we contacted numerous groups and none of them have identified a suitable replacement. However, given that this study would strongly support our conclusions, in the revision period we have attempted CEBPA ChIP-seq using a non-validated CEBPA antibody (CST, D56F10). Although we were able to generate libraries, the data obtained was not of high quality with only 300 high confidence peaks being called across replicates. The majority of these do align with published peaks from the 14AA antibody.

However, the sparsity of the peaks and variability between replicates preclude us making meaningful conclusions using this data. We also attempted CUT&RUN for CEBPA using the same antibody, but unfortunately were unable to generate libraries with sufficient yield for sequencing.

We therefore expanded our reanalysis of published ChIP-seq data for CEBPA performed in Granulocyte Macrophage Progenitors (GMPs) using the 14AA antibody. As Hoxb8 cells phenocopy GMPs, the pattern of CEBPA binding should be reflective of that which occurs in the absence of oncogenes. We found a strong statistical enrichment of CEBPA peaks in CSF3R^{T618I} specific enhancers consistent with the presence of CEBPA motifs at these enhancers. This provides supporting evidence for our hypothesis that activation of these differentiation-associated enhancers is dependent on wild type CEBPA and that mutant CEBPA disrupts this activation. We agree with the reviewer that the wording of our original conclusions would require ChIP-seq data across all oncogene conditions. We also agree that to truly understand the impact of mutant CEBPA, it will be crucial to distinguish between wild type CEBPA and mutant CEBPA. We believe that this will be best accomplished in the setting of endogenous CEBPA mutations rather than retroviral expression of mutant CEBPA in the presence of wild type, and are actively working along these lines.

To address this important point, we have made the following additions/changes:

- We have added an analysis of the publicly available ChIP seq data to Figure 5E, F.
- We have moderated our claims on lines 351-353 and also added a discussion of possible future validation experiments to lines 452-456.

“These data suggest that these differentiation-associated enhancers are regulated by direct CEBPA binding during normal hematopoiesis and the presence of mutant CEBPA prevents activation of these enhancers.”

“Going forward, it will be crucial to define the biological interaction with mutant CSF3R and N- and C-terminal mutations present in the endogenous to understand whether these mutant forms of CEBPA demonstrate differential binding or recruitment of cofactors to critical differentiation-associated enhancers.”

Response to Reviewer 2:

Braun et al. study the cooperation between CSF3R and CEBPA mutations in AML using retroviral transduction of mouse bone marrow cells and cell lines. Transcriptional (RNA-seq, microarray) and epigenetic (ChIP-seq) analysis informs genes and regulatory elements that are affected. It is convincingly demonstrated that CSF3R activation and CEBPA mutations cooperate to cause leukemia. Activating the JAK/STAT pathway and inhibiting CEBPA activity through other means has similar effects, which implies that various genetic lesions converge on a limited number of pathways in AML. It is interesting and to my knowledge novel that the order of these mutations determines the phenotype of the leukemia. For the most part, the work employs appropriate models and analyses, but I suggest numerous opportunities for improvements and additional controls below. Overall, the paper is a valuable contribution to the field of AML, with concepts that may apply to other tumors.

1. The relationship between AML-ETO, RUNX1, core binding factor AML and CEBPA is not common knowledge. It should be explained clearly, preferably in the introduction.

We agree that this component of the manuscript was not well described or validated experimentally. As discussed above, we have elected to remove data regarding AML-ETO from the manuscript to improve clarity and allow for additional space to discuss the fundamental findings.

2. Several comments about the paragraph “To further describe the transcriptional changes...”.

2.1 “Of particular interest was a cluster of genes that were strongly up-regulated by CSF3R-T618I, but suppressed by co-expression of CEBPA-V314VW, as this mirrored the pattern of myeloid differentiation.” The authors imply that CSF3R-T618I alone increases myeloid differentiation. Is that supported by functional assays in Figure 1?

We agree that this point is not well supported by the preceding figures. It is however supported by Figure 2C, Figure 4B, C and Supplemental Figure 4A. **We have also added morphologic characterization of the cells from colony assay to Figure 1E that support this conclusion.**

2.2 Figure 2D: Why are there eight heatmaps? Are they distinguished by K-means?

2.3 What is the rationale for calling the first cluster “interacting genes”?

2.4 How do the five columns in Figure 2E (and S1B) correspond to the eight heatmaps in Figure 2D? It may be clearer to use the same five categories in 2D.

Better description or heatmap of everything

We agree that this gene expression study was not well described and apologize for the confusion. The interacting genes are a statistical category where the cumulative effects of mutant CSF3R or CEBPA on expression are more or less than additive (i.e. synergistic or antagonistic). This can occur with a number of different expression patterns (i.e. induced by mutant CSF3R, unaffected by mutant CEBPA alone, but uninduced when both oncogenes are present in combination). The heatmap, clustered

by K-means into 8 groups, was an attempt to depict the varied expression patterns that occur within this interacting subset. We have made the following changes to clarify this analysis:

- We have modified the Figure panels 2A, B to clearly connect the interacting genes with the heatmap depicting our clustering analysis.
- We have added a more thorough description of this analysis to the results section on lines 169-173.

“Additionally, there were 570 genes that demonstrated an interaction between CSF3R^{T618I} and CEBPA^{V314VW} (effect less or more than additive). These interacting genes demonstrated a variety of patterns of regulation across all four oncogene conditions as demonstrated by K-means clustering (Figure 2B).”

We appreciate any further feedback regarding whether these changes have improved comprehensibility of this analysis and would be happy to make additional changes to the description to improve clarity.

2.5 “60% of the interacting genes were within 2 Kb of a CEBPA peak, compared with approximately 20% of non-interacting genes.” Please do this analysis for each of the five (or eight) categories. Considering you focus on enhancers later, 2 kb is a very small window.

As suggested by the reviewer, we have repeated this analysis with an extended window out to 10kb. We have also performed this analysis for each of the 5 categories of DE genes and found a similar association between CEBPA ChIP peaks and CEBPA down genes and CSF3R up genes along with the interacting subset. These results can be found in Supplemental Figure 1H.

3. “Interestingly, CSF3R mutant samples tended to display increased HOX gene expression although this did not rise to the level of statistical significance.” The legend of Figure 1F appears to be wrong. Assuming red is double mutant, it looks like there is no difference with the purple dots. This point can (and should) just be omitted.

We agree with the reviewer and have omitted this point as suggested

4. “Comparison of CEBPA mutant/CSF3R WT and CEBPA mutant/CSF3R mutant patient samples revealed numerous differentially-expressed genes (Table S2). Principle component analysis using these differentially-expressed genes clearly separates CEBPA/CSF3R mutant samples from those harboring mutant CEBPA alone (Figure S1D).”

It is “Principal”.

Thank you for catching this error, we have made the recommended change.

More consistency in naming groups would improve comprehensibility.

We would be happy to change the group naming strategy to improve clarity, but would appreciate more information on which group names are confusing so to make sure that we are correcting the right groups.

Most importantly, it is unsurprising that PCA using differentially expressed genes between two groups is going to separate those groups, this is circular.

We agree that it is unsurprising that the PCA analysis using DE genes is able to separate the groups. However, this is a quality control method to display sample to sample variability and also project the CEBPA mutational status which was requested by reviewer 3. We have elected to keep these plots in the manuscript as we do feel they provide some additional information. To address this issue, we have altered the text of the results to clarify the new information provided by the PCA plots on lines 226-227. As these plots are not crucial to the main point of the manuscript, we are amenable to removing them if the reviewer feels this would be a more appropriate strategy.

5. In Figure 3, the authors show in vivo cooperativity between JAK/STAT and CEBPA pathways. This is very interesting. It would be more convincing with additional controls. In 3A, why have the authors not done CEBPA-V314VW alone?

We appreciate the reviewer's enthusiasm for the interesting nature of our results. We agree that, as presented, it appears that controls were missing from Figure 3A. The CSF3R and the JAK3 experiments were done as a single experiment under identical conditions. The JAK/CEBPA transplant has the CEBPA alone control. We had separated this experiment into two subsets to improve readability of the figure. However, we now appreciate that interpretation of 3A requires a direct comparison with the CEBPA alone control. We have made the following changes to manuscript to address this point:

- We have combined the results of the mutant JAK and CSF3R transplant so that all data appears together with its appropriate control in Figure 3E-F.
- We have added an additional murine model in which the CSF3R^{T618I} mutation was added to endogenous CEBPA mutations (Figure 3 A-D) as described in detail in the response to reviewer 1. This experiment also contains all of the controls recommended by the reviewers and adds validation of our original findings.
- We have added a side by side histologic comparison of the leukemia that develops (or doesn't in the case of empty vector) with each combination of mutations to Figure 3G and Supplemental Figure 2.
- We performed an additional experiment designed to assess morphologic differences in the bone marrow of mice transplanted with either CSF3R^{T618I} or CSF3R^{T618I} + CEBPA^{V314VW} at a timepoint at which mice receiving the oncogene combination had developed AML but the CSF3R^{T618I} alone mice had not. This data was previously integrated with the above transplant data which made it unclear that it was a separate experiment. Therefore, we have separated this data into a new Figure (Figure 4) and added a description of the purpose of this experiment to the results section on lines 278-292.

- “However, to make a direct comparison, we transplanted syngeneic recipient mice with either 100,000 cells expressing CSF3R^{T618I} alone or 10,000 cells expressing CSF3R^{T618I} and CEBPA^{V314VW} (Figure 4A). This log-fold reduction in cell dose delayed the timing of disease onset to approximately 20 days, allowing for side by side comparison of these two groups. While CSF3R^{T618I} alone mice demonstrated an abundance of mature neutrophils (marked by high levels of CD11b and GR-1), the leukemic blasts seen in CSF3R^{T618I}/CEBPA^{V314VW} mice exhibited lower levels of GR-1 staining consistent with an immature myeloid phenotype (Figure 4B-D, Supplementary Figure 3D). Marked splenomegaly was observed in mice transplanted with CSF3R^{T618I}/CEBPA^{V314VW} cells but not in mice receiving CSF3R^{T618I} alone cells (Figure 4E-F). To establish whether the leukemic blasts generated by co-expression of CSF3R^{T618I} and CEBPA^{V314VW} were capable of indefinite self-renewal, we performed serial transplantation studies. These experiments revealed that the leukemia was capable of disease initiation in up to quaternary recipients (Supplementary Figure 3E-G).”

And why is the survival of CSF3R-T618I alone worse than the survival of CSF3R-T618I + CEBPA-F82fs?

We agree with the reviewer that this is a very interesting point and unfortunately do not have a complete explanation. We ultimately did not pursue a mechanistic investigation of this as the TARGET cohort only had one patient with mutant CSF3R and an N-terminal CEBPA mutation alone, so the disease relevance of this pairing is less clear. Based on the known biology, N-terminal mutations exert their oncogenic activity through cell cycle dysregulation while C-terminal mutations block differentiation. We suspect the latter is more important for CSF3R-mediated oncogenesis as mutant CSF3R also promotes proliferation. It may be that the proliferative impact of CSF3R and N-terminal mutant CEBPA produces an early acceleration of colony formation but ultimately is detrimental to the CSF3R mutant stem cell that drives later disease onset. As this explanation is purely speculative, we elected not to include in the manuscript.

In Figure 3D, what does a healthy control look like?

We agree that this is an important control. The data in this figure panel was from a separate experiment designed to investigate the morphologic differences between bone marrow cells harboring mutant CSF3R alone and the combination of mutant CSF3R and CEBPA as we were interested in understanding the differential oncogenic mechanisms present in CSF3R mutant CNL and CSF3R/CEBPA mutant AML. As discussed above we have restructured the retroviral bone marrow transplant studies into two separate figures. In Figure 3 and Supplemental Figure 2, 3 we present side by side comparisons to the leukemia that ultimately develops at the experimental endpoint and include for comparison purposes images and data from Empty/Empty controls and untransplanted mice. We have moved data comparing CSF3R^{T618I} alone to CSF3R^{T618I}/CEBPA at an early timepoint to Figure 4 to clarify that this represents a separate focused experiment.

In Figure 3K, what does AML-ETO alone look like?

We agree that this is an important control. Multiple other investigators have shown that retroviral transduction of mouse bone marrow with AML-ETO alone does not produce leukemia (Schesl et al, JCI, 2005). However, we agree that the finding of accelerated disease with the addition of mutant CSF3R cannot fully be supported in the absence of this control. As discussed above, we have elected to remove this data from the manuscript to improve clarity and rigor.

6. Figure S5D-S5E. The authors argue overlap between enhancers indicates conserved biology. How about a negative control, for example, there should be no enrichment when using human MLL AML enhancers or peaks.

To address this important point, we have included this analysis using all other AML samples in this dataset (i.e. CEBPA WT AML). We observe an enrichment in overlap with human CEBPA-Mutant enhancers and our murine leukemic enhancers as we reported in the previous version of the manuscript. Our expanded analysis demonstrates that human CEBPA-wild type enhancers demonstrate less overlap than expected by chance with our murine leukemic enhancers. This confirms that our epigenetic study is reproducing some key aspects of the human disease. These data can be found in Supplemental Figure 5A.

7. "We observed strong enrichment of CSF3R-T618I-specific enhancers in Clusters 1 and 4, which displayed the highest level of gene expression with CSF3R-T618I alone." Please add interpretation. It seems to me that Figure 4F shows that open enhancers are associated with increased gene expression.

We completely agree with this interpretation and this is precisely what we were trying to convey. We have revised the text to improve clarity. Please see lines 339-341.

"This analysis demonstrated globally that activated enhancers are associated with increased expression of the nearest gene."

8. The paragraph about enhancer priming and H3K4me1 intensity is difficult to understand. The first statement ("H3K4me1 enrichment ... CEBPA-V314VW only") is circular and the second statement ("Regions with specific enhancers") is not supported by any figure or statistical analysis. It would be better to omit this paragraph. The poorly supported conclusion that the mutations affect H3K4me1 is not vital to the paper. Figure 4H-I on the other hand are great.

We agree with the points made by the reviewer and have removed this section and data.

9. Figure 5E: Have the authors done CEBPA only?

We agree that ideally this would be a good control. Unfortunately, this experimental setup requires in vitro culture with tamoxifen in the absence of cytokines for 48 hours.

Only conditions with CSF3R^{T618I} are sufficiently viable at the end of this culture period for high quality RNA to be obtained. This is similar to what we see in colony assay where CEBPA mutants alone fail to support colony formation.

10. Figure 5G, Figure S6: Why did the authors pool results for CSF3R First + CSF3R Only? It seems a better comparison would be CEBPA First vs. CSF3R First.

We agree with the reviewer that this is a better way to present the data and have made this change and present this new comparison in Figure 6G

11. The last section about the order of mutations is interesting. Is it possible to compare AML patients with CSF3R VAF > CEBPA VAF to AML patients with CSF3R VAF < CEBPA VAF (classification, outcome, ...)?

We agree that this analysis would be very interesting. However, it is not clear that CEBPA VAF is particularly accurate as the GC rich nature of the gene precludes reliable detection with next generation sequencing methods. Indeed, the clinical assay in use at most centers is a Sanger sequencing based method, which is not robustly quantitative. Unfortunately, in the TARGET cohort a large fraction of CEBPA mutations were identified by Sanger not NGS and thus we do not have consistently reliable data on CEBPA VAF. For the TARGET patients with CSF3R/CEBPA mutant AML, outcome data will be presented soon at a national meeting, and, to date, does not seem to vary based on VAF.

12. Figure 7: It is great to have a graphical summary for clarification. However I find this figure confusing. Can you find a way to integrate some aspects of the attached figure?

Thank you for this suggestion. We have provided a simplified summary figure based on the reviewer's draft model. We appreciate any further feedback on ways to improve this new figure.

MINOR COMMENTS

13. In the abstract, it should be "applicable TO other".

Thank you for catching this, we have made the change.

14. Introduction: In the first sentence, "genetic" would be a better term than "genomic". CEBPA is "a", not "the", master regulator of myeloid lineage commitment. What is the frequency of CSF3R and CEBPA mutations in AML and the overlap?

Agreed, we have made these changes to lines 60 and 69 and added a discussion of the frequency of overlap to lines 87-88.

"Approximately 20-30% of patients with CEBPA mutant AML harbor a cooperating mutation in CSF3R."

15. Results: “sustained growth in liquid media lacking cytokines (Figure 1C).” That is not what Fig. 1C shows.

We agree this is not precisely what the figure shows. At the end of this replating, cells were placed in liquid culture and demonstrate sustained growth. There is not a great visual method of displaying this finding, therefore we have removed this statement to improve clarity.

16. The columns in Figure 2B-C-D should be in the same order.

The reanalysis of this dataset to directly compare mutant CSF3R alone vs mutant CSF3R plus mutant CEBPA as suggested by another reviewer was performed and precludes an assessment of these gene sets across all sample groups. These heatmaps have therefore been removed from the revised figures

17. As everybody knows, you have to describe figures in order. In the text, S1C is described before S1B. Similarly, Figure 2I is described before 2H. Similarly, Figure S5D-E are described before Figure S5A-C. Similar issues with Figure 6.

Thank you for catching this, these issues have been corrected

18. Discussion: “which no doubt alter the impact of RAS mutations acquired later in disease evolution” is a very strong statement.

We have replaced “no doubt” with “potentially” to soften the statement.

19. Figure S4C: Please add Empty Empty.

As discussed above, the data describing the interaction of AML-ETO and CSF3R have been removed from the manuscript.

20. Figure S5D: the legend is incomplete.

This has been corrected, thank you for catching this error.

21. Figure S7E: why does it say Week?

This error has been corrected

22. It is not clear which supplementary table is which. Please add the table name in first row.

As suggested, we have added table names in the first row.

Response to Reviewer 3:

The authors use retroviral over-expression in murine bone marrow cells to study interaction between CEBPA and CSF3R mutations in AML. They conclude that silencing of critical myeloid enhancers is important for the functional co-operation between these genetic lesions. Overall, this is an interesting study with an impressive amount of data. I have a number of specific comments.

Major comments:

1. The level of over-expression of the various CSF3R and CEBPA constructs in mouse bone marrow cells should be assessed by western blot (relevant to Figs 1 and 5). As the authors are using retroviral over-expression, these experiments are essential to interpret all the subsequent data. N-terminal CEBPA mutations typically result in expression of a truncated 30kDa isoform from an internal initiation site (PMID11242107) – can this be confirmed for the CEBPA^{F82fs} construct?

We agree with the reviewer that it is important to test that both constructs behave as predicted with respect to the p30 and p42 isoforms. To obtain a sufficient number of cells for an immunoblot analysis we transiently transfected a myeloid cell line without endogenous CEBPA expression (K562) with the CEBPA^{V314VW} and CEBPA^{F82fs} constructs. The immunoblot demonstrates that both the p42 and p30 isoform are expressed from the CEBPA^{V314VW} construct, but that there is a loss of p42 expression from the CEBPA^{F82fs} while expression of p30 is maintained. This data is now presented in Supplemental Figure 1A.

Additionally, to address the levels of expression in murine bone marrow cells we were able to use expanded bone marrow progenitor cells transduced by both CSF3R^{T618I} and CEBPA^{V314VW} (CEBPA^{F82fs} does not promote progenitor expansion, which is why we employed K562 cells to obtain enough cells to assess its banding patterns as described above). We identified high expression of both CSF3R and CEBPA relative to untransduced mouse bone marrow by immunoblot analysis. This data is now presented in Supplemental Figure 1A.

Likewise, the impact of mutant CSF3R on downstream signaling should be analyzed.

Membrane proximal CSF3R mutations are known to activate down-stream JAK/STAT and MAPK signaling (Maxson et al, JEM, 2013). To confirm that this also occurs in the context of mutant CEBPA, we performed a western blot on CSF3R^{T618I} CEBPA^{V314VW} mutant leukemia cells for pSTAT3 and pERK which confirms activation of these pathways compared with normal bone marrow. These results now appear in Supplemental Figure 1G.

2. In many published patient cohorts, bi-allelic CEBPA mutations are more frequent than mono-allelic mutations and have distinct transcriptional, epigenetic and prognostic features (e.g. PMIDs 27276561 and 28408400). Collectively there is a large body of work that argues that CEBPA mutations are functionally distinct from loss of CEBPA expression. It is not entirely clear what the authors are attempting to model with their over-expression strategy (presumably mono-allelic mutations). To their credit, they do

attempt to reconcile gene expression data from their model with data from AML patients. However, I would suggest:

(1) the issue of mono-allelic vs bi-allelic CEBPA mutations should be dealt with in the introduction;

We agree that this is a crucial issue. We further agree that biallelic CEBPA mutations have unique biologic characteristics distinct from monoallelic cases. However, the association between CSF3R mutations and CEBPA mutations does not appear to be restricted to biallelic cases either in pediatric or adult AML (Maxson et al, Blood, 2016, Lavallée et al, Blood, 2016). In the pediatric cases, all but one patient with both mutant CSF3R and CEBPA had a C-terminal mutation, while the N-terminal mutation was found only in 2/3 of patients.

Given that this issue is so important, we have elected to include new data modeling the combination of biallelic CEBPA mutations with mutant CSF3R.

- Through a collaboration with Claus Nerlov, we have obtained fetal liver hematopoietic cells harboring a compound heterozygous N and C terminal CEBPA mutation (Bereshchenko et al, Cancer Cell, 2009). When transplanted into syngeneic recipients these cells produce a myeloid leukemia over the course of approximately one year. Prior to transplantation, we transduced these fetal liver cells with mutant CSF3R. This experiment had been initiated prior to our initial submission, however owing to the long latency of disease development we did not include it with our first submission. However, it has sufficiently matured at this point and we are now pleased to include these data. They confirm our finding that mutant CSF3R accelerates disease development in the context of endogenous compound heterozygous CEBPA mutation (Figure 3A-D)

(2) the caveats of the modelling approach should be outlined in the discussion; and

We have updated the introduction and discussion to contain a more complete discussion of the pattern of mutation co-occurrence, our new biallelic CEBPA mutant data and the caveats of this model. Text changes are present on lines 69-84 and 433-456.

“The transcription factor CCAAT enhancer binding protein alpha (CEBPA) is a master regulator of myeloid lineage commitment. CEBPA is recurrently mutated in AML and is a classic example of a Class II mutation². The CEBPA gene is comprised of a single exon with an internal translational start site with mutations clustering at the N- and C-terminus of the protein. N-terminal mutations typically result in a frame shift, leading to a premature stop codon and loss of expression of the long (p42) isoform of the protein but ongoing translation of the short isoform (p30). The p30 domain lacks a crucial transactivation domain that represses cell cycle progression through a direct interaction with E2F³. In contrast, C-terminal mutations occur in the DNA binding domain leading to loss of function and a blockade in granulocytic differentiation². The most common pattern of mutation in AML is an N-terminal mutation on one allele and a C-terminal mutation on the other allele. This results in alterations in the ratio of the p42 to p30 CEBPA isoforms, changing the balance of differentiation and proliferation. AML with

biallelic CEBPA mutations is associated with favorable prognosis, with approximately 50% of younger patients achieving a cure with chemotherapy alone. However, the precise determinants of relapse in this mutational context are unknown.”

“N- and C-terminal CEBPA mutations exert distinct biological roles in leukemia initiation^{21,28}. While it is clear that reduced CEBPA activity is potentially oncogenic, CEBPA knockout mice fail to develop AML²⁹. Similarly, mice with homozygous C-terminal mutations (putatively loss of function) develop leukemia with an exceedingly long latency²¹. Interestingly, the phenotype of these leukemias is erythroid rather than myeloid. This is consistent with data from MLL-rearranged leukemias, where CEBPA is requisite for entrance into the myeloid lineage and leukemia initiation^{30,31}. As the majority of patients with CEBPA mutant AML harbor combined N- and C-terminal CEBPA mutations, it is possible that the N-terminal CEBPA mutation provides sufficient residual myeloid differentiation potential to enter the myeloid lineage and initiate AML. This relationship becomes even more complex when CSF3R mutations are considered. While the majority of CSF3R mutations occur in CEBPA-bi cases there are a number of CSF3R mutant CEBPA C-terminal monoallelic cases as well⁵. Our data demonstrate that CSF3R^{T618I} potentially synergizes with biallelic CEBPA mutations to induce AML. Our retroviral model best represents monoallelic cases (with the caveats of ectopic expression), and demonstrates that C- but not N-terminal CEBPA mutations exhibit synergy with mutant CEBPA confirming the high degree of clinical association between these mutations. It is possible that mutant CSF3R provides both a myeloid commitment signal and proliferative advantage, thus rendering N-terminal CEBPA mutations less crucial for AML initiation. Going forward, it will be crucial to define the biological interaction with mutant CSF3R and N- and C-terminal mutations present in the endogenous to understand whether these mutant forms of CEBPA demonstrate differential binding or recruitment of cofactors to critical differentiation-associated enhancers.”

(3) when analysing patient data, it should be clarified whether all CEBPA mutant patients are being considered together (or whether mono-allelic and bi-allelic patients are considered as separate entities).

We have updated the results section to more clearly describe the analysis we performed on patient data. Given the small number of patient samples harboring mutant CEBPA, we elected to include all CEBPA mutant patients irrespective of whether they were biallelic or mono allelic, but have now added annotations as to the mutation types in each case. Of the 9 patients described, 7 were biallelic and 2 were monoallelic. The clarifying text changes can be found on lines 212-213. The clinical annotation for these samples can be found in Supplemental Table 2. For the differential expression analysis, CEBPA mutational status for the CSF3R/CEBPA mutated patients is indicated on the PCA plot in Figure 2 H, J).

“Given the small number of patients, both monoallelic and biallelic CEBPA mutations were considered together.”

3. The authors propose a model whereby mutant CEBPA displaces wild-type CEBPA from enhancers formed downstream of CSF3R mutation (Fig.7), however, this is largely based on motif enrichment. Direct evidence for this proposed model by way of CEBPA ChIPseq (or at least ChIP-PCR at a panel of target enhancers) would strengthen the study (and should be achievable in HoxB8 cells).

We appreciate the reviewer's suggestion and agree that CEBPA ChIP seq would be a valuable addition to the manuscript. Unfortunately, the CEBPA antibody used for ChIP seq by the majority of labs (Santa Cruz 14AA) has been discontinued. At the outset of this project we contacted numerous groups and none of them have identified a suitable replacement. However, given that this study would strongly support our conclusions, we have attempted CEBPA ChIP seq using a non-validated CEBPA antibody (CST, D56F10). Although we were able to generate libraries, the data obtained was not of high quality with only 300 high confidence peaks being called across replicates. The majority of these do align with published peaks from the 14AA antibody. However, the sparsity of the peaks and variability between replicates preclude us making meaningful conclusions using this data. We also attempted CUT&RUN for CEBPA using the same antibody, but unfortunately were unable to generate libraries with sufficient yield for sequencing.

We therefore did a reanalysis of published ChIP-seq data for CEBPA performed in Granulocyte Macrophage Progenitors (GMPs) using the 14AA antibody. As Hoxb8 cells phenocopy GMPs, the pattern of CEBPA binding should be reflective of that which occurs in the absence of oncogenes. We found a strong statistical enrichment of CEBPA peaks in CSF3R^{T618I} specific enhancers consistent with the presence of CEBPA motifs at these enhancers. This provides supporting evidence for our hypothesis that activation of these differentiation-associated enhancers is dependent on wild type CEBPA and that mutant CEBPA disrupts this activation. We agree that the strength of our original conclusions was not strongly supported by the data and would require ChIP seq data across all oncogene conditions. We have therefore moderated our claims and added a discussion of future directions.

To address this important point, we have made the following additions/changes:

- We have added an analysis of the publicly available ChIP seq data to Figure 5E, F
- We have moderated our claims on lines 351-353 and added a discussion of possible future validation experiments to lines 452-456.

“These data suggest that these differentiation-associated enhancers are regulated by direct CEBPA binding during normal hematopoiesis and the presence of mutant CEBPA prevents activation of these enhancers.”

“Going forward, it will be crucial to define the biological interaction with mutant CSF3R and N- and C-terminal mutations present in the endogenous to understand whether

these mutant forms of CEBPA demonstrate differential binding or recruitment of cofactors to critical differentiation-associated enhancers.”

4. Related to the above point, I am unclear how the model explains the functional cooperation between CSF3R and CEBPA mutations. It partially explains the block of differentiation (failure to activate enhancers important for differentiation) but if wild-type CEBPA is required for mutant CSF3R activity then CEBPA loss of function and CSF3R gain of function mutations would be antagonistic. Are there genes/enhancers that are uniquely activated in the CSF3RT618I/CEBPAV314VW double mutant cells that may explain the phenotype? Or is it perhaps that the differentiation-associated CSF3RT618I-gene expression programs are CEBPA-dependent but the proliferation-associated programs are not? I think some more detail is required here. Functional validation of key target genes may also be appropriate.

We completely agree that differentiation-associated programs downstream of mutant CSF3R depend upon wild type CEBPA but proliferative programs do not. Our focus in this manuscript was to describe the mechanistic underpinnings that separate the phenotypic differences between CSF3R mutant chronic neutrophilic leukemia (no CEBPA mutation) and AML (CEBPA mutation). However, we agree that the proliferative aspect requires additional detail and validation.

- Our gene expression analyses identified Myb as a key transcriptional target of mutant CEBPA. Myb is a known driver of cellular proliferation. We now present functional validation data showing that overexpression of Myb augments the colony output in response to mutant CSF3R in Supplemental Figure 1D, E.
- We performed a GO analysis on enhancers that were activated by mutant CSF3R irrespective of the presence of mutant CEBPA. This analysis revealed a strong enrichment for cell cycle and proliferative pathways. An example enhancer at the E2F2 locus is now shown in Figure 5H. Expression of this gene is strongly induced by mutant CSF3R with or without mutant CEBPA. Overlay of published Stat3 ChIP seq data demonstrates a clear Stat3 binding site at this enhancer consistent with E2F2 being a direct CSF3R target gene.

Minor comments:

1. The authors compare active enhancers identified specifically in CSF3RT618I/CEBPAV314VW HoxB8 cells with enhancers in CEBPA mutant patients and find extensive overlap (Figs S5D and S5E). This seems to go against their argument that CSF3R and CEBPA mutations co-operate functionally to alter the enhancer landscape in a manner that is more than additive. Can the authors please provide an explanation?

We apologize for the confusion regarding this analysis. We added this analysis to correlate the findings in our murine system with existing human epigenetic data. In our system, mutant CEBPA alone produces differentiation arrest but fails to produce AML in the majority of mice. In cases where AML does evolve, it does so with long latency suggesting the possibility that additional genetic events are necessary for disease initiation. Co-expression of mutant CSF3R rapidly initiates AML and produces a cell line

capable of indefinite growth in liquid culture. Thus, the combination of mutant CSF3R and mutant CEBPA are sufficient to produce AML. Therefore, we felt that the best comparison would be between our leukemia specific enhancers (i.e. those activated only in the presence of CSF3R^{T618I}/CEBPA^{V314VW}) with those found in fully developed human AML where multiple genetic drivers are present (Mutant CEBPA plus presumably a second cooperating mutation as unfortunately this cohort was limited in size and does not contain reported CSF3R mutations). This analysis is limited, but we do feel that it does provide some support for the notion that our murine system is recapitulating some features of human AML.

We would appreciate further feedback from the reviewer if it would be better if this analysis were omitted.

2. The authors use an elegant system to test the importance of mutation order for the phenotype and conclude that “when CEBPA mutations are introduced after mutations in CSF3R, they are unable to fully block myeloid differentiation”. To fully justify their conclusion the authors would need to show (e.g. by FACS) that at the time when they induce mutant CEBPA expression in CSF3RT618I-first cells, the cells are not already differentiated (relative to an empty vector control). I think that a more likely explanation is that that loss of CEBPA activity cannot reverse differentiation.

We appreciate the reviewer’s enthusiasm for our system to control mutation order. We agree that this point is crucial for interpretation of our results. For the RNA sequencing studies in Figure 2 and Figure 6, we transduced whole bone marrow but then FACS sorted lineage negative bone marrow transduced with both oncogenes (GFP and RFP). We did consider the approach suggested by the reviewer, however it is well established that progenitors differentiate in ex vivo culture and we were concerned that our mutations would influence this differentiation during the 3 days of culture necessary for efficient retroviral transduction. We also considered transducing whole bone marrow and sorting cKit positive progenitors receiving both oncogenes. However, we found that cKit expression is rapidly lost during the necessary 3-day culture. Thus, we felt that sorting lineage negative cells was the best way to control perturbations in the mixture of cells as the reviewer points out. Interestingly, rather than observe a decrease in the percentage of lineage negative cells, we found that **mutant CSF3R actually drove an increase in the size of the lineage negative population**, consistent with promoting stem cell proliferation.

- We have added figure panels clearly depicting our flow sorting strategy to Supplemental Figure 1B, C and Supplemental Figure 6A, B.

3. Related to the above point, analysis of enhancers in the sequential model (e.g. ChIP for H3k27ac in the CEBPA-first cells versus the CSF3R-first cells) could prove really informative

We agree that this would be an exciting and informative analysis. One technical challenge with such an analysis is that cell yields are low from sorts for lineage negative cells expressing both oncogenes. While methods exist to profile covalent histone

modifications from 10-20,000 cells, these are complex and not optimized for primary bone marrow cells. We are in the process of optimizing such methods, but this process is still in the early stages.

4. Figure 4C does not seem to be referenced in the text.

We apologize for this omission. We have now included a reference to this figure.

5. There appears to be something wrong with the PCA plot in Figure S5C with no variation between samples in the first principle component

We apologize for the confusion regarding this PCA plot. It is our convention to perform PCA analysis with genomic regions as rows. In this setting, genomic position biases such as GC content and blacklisted regions numerous (hence the need for an input control). These effects generally tend to dominate PC1. The stacking of the samples in this setting indicate that these biases are similar between samples with sample to sample variation appearing in PC2. As we agree that this analysis is confusing, we have omitted it from the text and added brief mention to the QC analysis to the methods on lines 667-668.

“For quality control purposes, PCA was performed using Deeptools⁵⁰ which showed separation of signal by oncogene condition.”

6. Scales/labels are missing on some figures:

- Fig 5E: scale label
- Fig S1A: missing key for point size (p value?)
- Fig S5A: scale labels, y axis metagene plot label missing

We apologize for the omission and have added these legends.

7. “morphologic neutrophils in the peripheral blood and bone marrow (Figures 2A-B...)” I think is referring to Figures S2A-B)

We apologize for this error and have correct the reference to the appropriate figure.

REVIEWERS' COMMENTS:

Reviewer #1 (Remarks to the Author):

The authors responded very adequately to my comments and suggestions. Based on this and on the response to the other reviewers they have improved the manuscript such that I find it acceptable for publication in Nature Communications.

Reviewer #2 (Remarks to the Author):

The revised manuscript is significantly better with improvements in structure, clarity, controls and support for the conclusions.

Line 168: "809 genes": I only see 773 in Figure 2A.

Line 219, "numerous": specify

Line 225: "preponderance": what percentage?

Line 361: "One likely driver of this CSF3R-induced proliferation is the transcription factor E2f2 which is strongly induced by CSF3RT618I in the presence and absence of CEBPAV314VW (Figure 5H)". Please clarify, I only see a small difference in H3K27ac.

Figure 2C/D: The text below the heatmap is unclear (which genotype belongs to which column).

Figure 2H/J arrows: What does "CEBPA BI" mean, and which gene do the mutations in J refer to?

Figure 8: Is the second panel necessary or add unnecessary complexity? Make the legend as clear as possible, perhaps including a reminder that CSF3R mutations drive proliferation and CEBPA mutations block differentiation.

Peter van Galen

Reviewer #3 (Remarks to the Author):

The authors have made efforts to address my comments and have significantly improved the manuscript. Some specific comments:

(1) The lack of ChIPseq for CEBPa (in the presence or absence of CSF3RT618I and/or CEBPAV314VW) continues to be a limitation of the paper. The lack of a suitable antibody for CEBPA ChIPseq could be overcome by other methodologies (e.g. knocking in a FLAG tag into the endogenous Cebpa locus in HoxB8 cells or using over-expression of FLAG-tagged CEBPa), however, these experiments may be beyond the scope of the current work.

(2) I found Figure 8 somewhat confusing. Panel 1 and 2 have the same graphic (with respect to cell numbers) even though 2 is supposedly representing Chronic Neutrophilic Leukemia. The AML situation is also not represented in any of the panels

Response to Reviewers Comments:

REVIEWERS' COMMENTS:

Reviewer #1 (Remarks to the Author):

The authors responded very adequately to my comments and suggestions. Based on this and on the response to the other reviewers they have improved the manuscript such that I find it acceptable for publication in Nature Communications.

Reviewer #2 (Remarks to the Author):

The revised manuscript is significantly better with improvements in structure, clarity, controls and support for the conclusions.

Line 168: "809 genes": I only see 773 in Figure 2A.

We appreciate the reviewer careful attention to detail. Reviewing the original data, there are only 773 genes differentially expressed in response to the expression of mutant CEBPA. We have confirmed the other numbers listed and they are correct.

Line 219, "numerous": specify

913 genes are differentially expressed. This has been added to the text

Line 225: "preponderance": what percentage?

The exact percentage (68%) has been added to the text.

Line 361: "One likely driver of this CSF3R-induced proliferation is the transcription factor E2f2 which is strongly induced by CSF3RT618I in the presence and absence of CEBPAV314VW (Figure 5H)". Please clarify, I only see a small difference in H3K27ac.

The enhancer region in question was based on a MACS2 peaks called only in the presence of CSF3R-T618I. We agree that this is visually less impressive due to the necessary scaling to accommodate the larger peak at the gene promoter. However, this region does display statistically significantly differential H3K27Ac.

Figure 2C/D: The text below the heatmap is unclear (which genotype belongs to which column).

We have changed the labels to vertical orientation to improve clarity.

Figure 2H/J arrows: What does "CEBPA BI" mean, and which gene do the mutations in J refer to?

We have added text to the figure legend to clarify these labels

"Cases with biallelic CEBPA mutations are indicated by CEBPA-Bi I. Expression of genes differentially expressed in both murine and adult human CSF3R/CEBPA AML as compared with CEBPA-mutant CSF3R-WT AML. J. PCA analysis of adult

CEBPA mutant AML using convergent human-mouse gene set. Specific CEBPA mutations are indicated by the text.”

Figure 8: Is the second panel necessary or add unnecessary complexity? Make the legend as clear as possible, perhaps including a reminder that CSF3R mutations drive proliferation and CEBPA mutations block differentiation.

We appreciate the feedback regarding the summary figure. We have removed panel 2 added a reminder that CSF3R mutations drive proliferation and CEBPA mutations block differentiation as suggested.

Peter van Galen

Reviewer #3 (Remarks to the Author):

The authors have made efforts to address my comments and have significantly improved the manuscript. Some specific comments:

(1) The lack of ChIPseq for CEBPa (in the presence or absence of CSF3RT618I and/or CEBPAV314VW) continues to be a limitation of the paper. The lack of a suitable antibody for CEBPA ChIPseq could be overcome by other methodologies (e.g. knocking in a FLAG tag into the endogenous Cebpa locus in HoxB8 cells or using over-expression of FLAG-tagged CEBPa), however, these experiments may be beyond the scope of the current work.

We agree that the genome wide assessment of CEBPA binding is an important extension of our work. Although these studies could be performed using an epitope tag, we believe they would be best evaluated in the context of mutations in the endogenous allele (i.e. K/L mouse model). Therefore, we are evaluating a panel of CEBPA antibodies as well as alternate profiling methodologies such as CUT&RUN which could potentially allow us to address this question. However, we also agree that these additional experiments are beyond the scope of the current work.

(2) I found Figure 8 somewhat confusing. Panel 1 and 2 have the same graphic (with respect to cell numbers) even though 2 is supposedly representing Chronic Neutrophilic Leukemia. The AML situation is also not represented in any of the panels

We appreciate the feedback regarding the summary figure. We have removed Panel 2 and clarified that the new panel 2 represents the AML situation.